# Early response evaluation by single cell signaling profiling in acute myeloid leukemia

Benedicte Sjo Tislevoll [1], Monica Hellesøy [2], Oda Helen Eck Fagerholt [1], Stein-Erik Gullaksen[2], Aashish Srivastava [3], Even Birkeland[4], Dimitrios Kleftogiannis [1,5], Pilar Ayuda-Durán [6,7], Laure Piechaczyk[6,7,8], Dagim Shiferaw Tadele [9,10], Jørn Skavland[1], Panagotis Baliakas [11], Randi Hovland [12], Vibeke Andresen [1], Ole Morten Seternes [13], Tor Henrik Anderson Tvedt[14], Nima Aghaeepour [15,16,17], Sonia Gavasso[1,18], Kimmo Porkka [19], Inge Jonassen [5], Yngvar Fløisand [7,14], Jorrit Enserink [6,7,20], Nello Blaser [21] ✉ & Bjørn Tore Gjertsen [1,2] ✉

Aberrant pro-survival signaling is a hallmark of cancer cells, but the response to chemotherapy is poorly understood. In this study, we investigate the initial signaling response to standard induction chemotherapy in a cohort of 32 acute myeloid leukemia (AML) patients, using 36-dimensional mass cytometry. Through supervised and unsupervised machine learning approaches, we find that reduction of extracellular-signal-regulated kinase (ERK) 1/2 and p38 mitogen-activated protein kinase (MAPK) phosphorylation in the myeloid cell compartment 24 h post-chemotherapy is a significant predictor of patient 5-year overall survival in this cohort. Validation by RNA sequencing shows induction of MAPK target gene expression in patients with high phospho-ERK1/2 24 h post-chemotherapy, while proteomics confirm an increase of the p38 prime target MAPK activated protein kinase 2 (MAPKAPK2). In this study, we demonstrate that mass cytometry can be a valuable tool for early response evaluation in AML and elucidate the potential of functional signaling analyses in precision oncology diagnostics.

Acute myeloid leukemia (AML) is a hematopoietic stem cell-derived myeloid malignancy characterized by manifold genetic aberrations and poor overall survival[1–3]. Standard treatment for newly diagnosed AML patients fit for intensive chemotherapy includes a combination of an anthracycline (daunorubicin 60–90 mg/m² or idarubicin 10–12 mg/m²) for three days and cytarabine (Ara-C) (200 mg/m²) for seven days with an initial remission rate of 60–80%[2]. However, relapse is a challenge in more than 40% of AML patients. In AML, like in most cancers, response to therapy is evaluated weeks to months after the start of therapy. Early detection tools of responders and non-responders will be essential to improve cancer patient survival, providing physicians with patient-specific information to change treatment strategy early and avoid unnecessary adverse effects[4–7].

Aberrant signaling in cancer is known to regulate cancer cell proliferation, protect against cell death and modulate interactions with the micro-environment[8]. Approximately 60% of AML patients harbor mutations in signal transduction pathways[9], including mutations resulting in abnormal activation of signaling that is associated with prognosis[2]. Conventional chemotherapeutics affect signaling directly or through the induction of cell stress. In AML, this is demonstrated within hours after the start of chemotherapy, involving gene expression of proteins central in the regulation of cell death and survival[10,11].

The intratumor heterogeneity of AML underscores the need for methods that provide single-cell resolution. Single-cell immune and signaling profiling by mass cytometry provides a high dimensional

method that deciphers the phenotypic- and functional heterogeneity in cancer by simultaneous analysis of multiple parameters in single cells[12]. The application of >40 antibodies permits simultaneous analysis of both intracellular signaling networks and phenotypic characterization of cells[13]. Furthermore, the possibility of sample multiplexing allows for direct comparison of sequentially acquired patient samples, making this an ideal tool for the assessment of therapy response in a clinical setting[14].

The current risk stratification for predicting long-term response in AML is based on genetic analyses in cancer cells sampled at the time of diagnosis[2]. However, risk stratifications that include clinical and biological information are under discussion[15]. Functional analyses of drug sensitivity and signaling responses are suggested[13,16–18], but based on our previous observations of protein and gene expression hours after the start of chemotherapy[10,11], we hypothesize that early chemotherapy-induced alterations in intracellular signaling networks may be a more accurate predictor of long-term therapy response.

Here, we employ mass cytometry to investigate intracellular signaling networks in peripheral blood (PB) samples from 32 newly diagnosed AML patients during the first 24 h of standardized induction chemotherapy. By correlating initial intracellular signaling response to 5-year overall survival, we demonstrate that early response evaluation by mass cytometry at 24 h identifies patients with suboptimal treatment response to standard induction therapy.

## Results

### Characterization of major immunophenotypic clusters in AML by FlowSOM

Based on experience with proteomics-based functional diagnostics in leukemic patients[11,19,20], we collected PB samples for single-cell signaling profiling from 32 AML patients immediately before, at 4 h and 24 h after the start of standard "7 + 3" induction chemotherapy (Fig. 1, Supplementary Fig. 1, Supplementary Tables 1 and 2, Supplementary Data 1 and 2). These patients were risk classified by ELN 2017 risk classification and assessed by conventional response evaluation by BM aspiration at day 17 post-treatment or before cycle two of induction therapy. (Fig. 1b, d). The antibody panel of 21 extracellular markers (Supplementary Table 3, Supplementary Data 3) was designed to identify the major immunophenotypic cell subsets in the samples, including both healthy immune cells and leukemic blasts. The inclusion of 15 intracellular markers allowed the investigation of the major intracellular signaling pathways regulating myeloid cell proliferation and survival[19,21]. To avoid user bias and enable robust and efficient analysis of our dataset, we applied the unsupervised clustering algorithm FlowSOM[22] (Fig. 1a).

AML is a disease with a large degree of intra- (and inter-) individual immunophenotypic heterogeneity. This is a challenge in multi-parameter flow cytometry or mass cytometry analysis because strict clustering based on surface marker expression leads to the identification of multiple unique blast clusters, often represented in only one or a few patients. In this work, we chose an analytical approach of under-clustering the data to capture the immature myeloid cell compartment in as few clusters as possible, while simultaneously identifying the major (presumably) healthy cell subsets. Manual analysis of increasing resolutions from 1 to 20 metaclusters (MCs) demonstrated that a total of 10 MCs was sufficient to differentiate the major healthy cell populations in PB from the AML blast cells (Fig. 2a). The 10 FlowSOM-identified MCs were annotated manually based on surface marker expression (Fig. 2c). Healthy cell types (MC3–8) had a homogeneous surface marker expression across patients and healthy donors (Supplementary Fig.2). The AML blast clusters (MC1 and MC2), the myeloid cluster (MC9) and the hematopoietic stem cell cluster (MC10) were, as expected, more heterogeneous. MC1 and MC2 had a high expression of CD34 and CD117 and were therefore classified as CD34+ blast cells (Fig. 2c and Supplementary Fig. 2). Most patients in this

cohort had a CD34+CD117+ blast population present in MC1, whilst MC2 was nearly entirely composed of patient 9 and 22, having mutated FLT3-ITD and WT1 (Fig. 2b and Supplementary Fig. 1). MC2 could primarily be distinguished from MC1 by higher expression of HLA-DR, CD33, and CD123. MC9 included most cells, with 43.2% of the cells in the entire cohort, and was represented in all patient samples and healthy donors. MC9 had a myeloid phenotype (CD64 dim, CD33 dim, and HLA-DR dim) and did not express lymphoid markers. This cluster had heterogeneous CD34 expression among patients and was negative for CD117 (Supplementary Fig. 2). MC9 was expanded in AML patients and bone marrow (BM) of healthy donors but not in PB of healthy donors. (Fig. 2b and Supplementary Fig. 3).

The healthy donors had a similar distribution of different cell populations that resembled a normal differential blood count, with approximately 60% granulocytes, 30% lymphocytes, and 3% monocytes. The AML patients showed an abnormal distribution of cell populations, with expanded myeloid lineage and CD34 + cells, while expansion of these cells was not detected in healthy donors (MC1/MC2) (Fig. 2b).

Overall, the size of the MCs was quite consistent between the different time points (Supplementary Fig. 4). However, there was a tendency among several patients towards a more differentiated phenotype at 24 h, with a decrease in the myeloid cluster MC9 and an increase in granulocytes (MC7) and monocytes (MC6) (Supplementary Fig. 4). There was no significant association between MC size and patient survival or ELN 2017 risk class, neither at the pre-treatment time point nor over the time course.

### Chemotherapy-induced changes in intracellular signaling during the first 24 h of treatment may predict long-term survival

Chemotherapy modulates intracellular phospho-proteins, including p53 and ribosomal P2, in bulk samples of tumor cells[11,20]. Therefore, we examined the treatment-induced phospho-signaling in each MC in the patient samples. We employed a machine-learning approach using a supervised LASSO Cox regression model with automated feature selection and nested leave-one-out cross-validation to evaluate which features were predictive of patient survival (Supplementary Fig. 5). Features evaluated in the model included age, sex, the 10 MC sizes (% of total), in each MC the 90th percentile dual count measured for all the functional markers (cCaspase3, CyclinB1, p-4E-BP1(T37/T46), p-AKT(S473), p-AXL(Y779), p-CREB(S133), p-ERK1/2(T202/Y204), p-Histone3(S28), p-NF-kB p65(S529), p-p38(T180/Y182), p-Rb(S807/S811), p-S6(S235/S236), p-STAT1(Y701), p-STAT3(Y750), and p-STAT5(Y684)) at all three timepoints (pre-treatment, 4, and 24 h), and in all MCs the change in all functional markers from pre-treatment to 24 h (ratio 24 h). The analysis identified p-ERK1/2 ($p = 2.057e-4$, p-adj = 0.0004, Log-hazard ratio (Log-HR) 2.41) at 24 h in MC9 as the most significant predictor for patient 2-year overall survival (2y-OS) (Fig. 3a, b). Furthermore, we repeated the LASSO analysis with 5-year overall survival (5y-OS), which revealed that p-ERK1/2 ($p = 9.59e-5$, p-adj = 0.0003, Log-HR 2.51) at 24 h in MC9 still was the most significant feature (Fig. 3c). When the cohort was split by median p-ERK1/2 at 24 h, resulting in 16 patients in each arm (Fig. 3d), we found that the patients with a high level of p-ERK1/2 in MC9 at 24 h after the start of induction chemotherapy had significantly inferior survival ($p = 0.0049$ 2y-OS, $p = 0.0015$, 5y-OS) compared to patients with p-ERK1/2 levels below median at 24 h (Fig. 3c). These two arms of 16 patients each will further be referred to as the high and the low 24 h-p-ERK1/2 groups, respectively. p-ERK1/2 levels at pre-treatment or 4 h after the start of induction therapy were not significant in predicting patient 5y-OS. The 24 h sample gave a better separation of deceased and alive patients (Supplementary Fig. 6a), and the 24 h levels of p-ERK1/2 in the 24 h-pERK1/2 high group were significantly higher than the levels in PB and BM from healthy donors (Supplementary Fig. 6b). To exclude potential confounding factors, we performed a new Cox regression analysis of

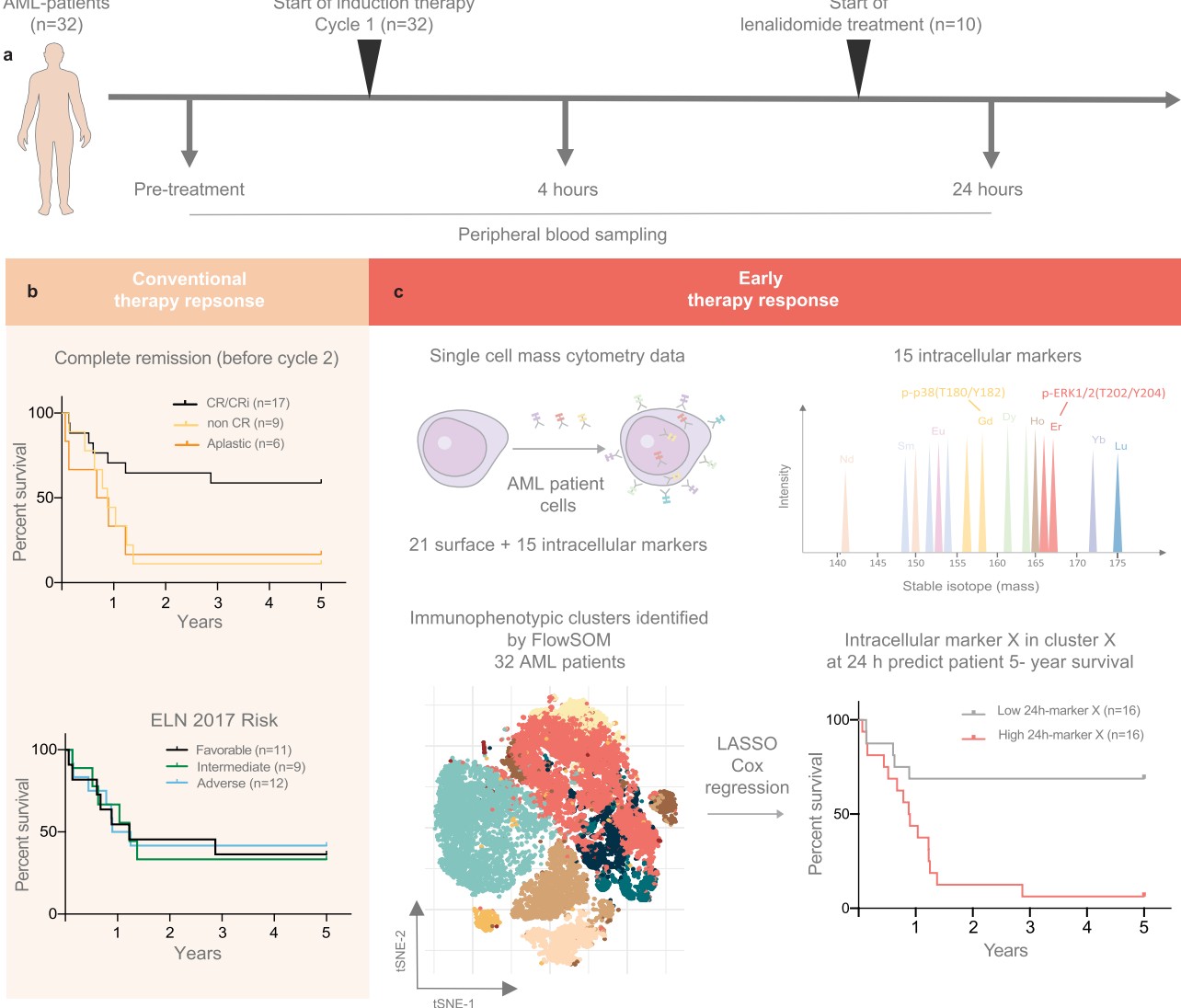

**Fig. 1 | Mass cytometry analysis of early response evaluation by single-cell signaling to profile. a** Peripheral blood samples were collected from 32 AML patients treated with conventional induction therapy ("7 + 3" cytarabine + daunorubicin). Samples were collected before the start of treatment, at 4- and 24-h after the start of treatment, and immediately fixed to preserve in vivo signaling. Ten patients in this study received per-oral treatment of lenalidomide in addition to the 7 + 3 induction therapy from days 1–21. **b** Five-year survival Kaplan–Meier curves showing the survival for the 32 AML patients in this study based on conventional therapy response assessment and European Leukemia Net (ELN) 2017 risk classification. Conventional therapy response assessment was done by bone marrow aspiration on day 17 post-treatment or before cycle two of induction therapy. Seventeen patients had CR/CRi, nine patients had nonCR and six patients were aplastic before the second cycle of induction therapy. Based on the ELN 2017 Risk classification, 11 patients had favorable risk, nine had intermediate risk and 12 had adverse risk. **c** Early therapy response assessment by mass cytometry at 4- and 24-h post-treatment by investigation of intracellular signaling response to chemotherapy. Machine learning approaches were used to identify markers in the blast cell population that could be predictive of patients' 5-year survival hours after the start of induction therapy (Kaplan–Meier curve, 16 patients in each group). Source data are provided as a Source Data file.

pERK1/2 in MC9 and included the validated prognostic factors of ELN 2017 risk, age, WBC at time of diagnosis and allogeneic stem cell transplantation as a time-dependent covariate. The pERK1/2 value at 24 h in MC9 was the only predictive marker for the patient outcome (5-yOS, $p$-value 0.000581, p-adj = 0.0029, Log-HR 2.27).

By further inspection of the longitudinal signaling patterns from pre-treatment to 24 h, a substantial drop in p-ERK 1/2 was observed in the low 24 h-p-ERK1/2 groups and vice versa in the high 24 h-p-ERK1/2 groups (Fig. 3d). We, therefore, calculated the ratio and delta between pre-treatment and 24 h for p-ERK1/2 and performed a univariate survival analysis by dividing the patients into two groups by the median. Both p-ERK1/2 ratio and delta significantly discriminated 5y-OS ($p$ = 0.0333) (Supplementary Fig. 6c, d). This confirms that 24 h of chemotherapy-induced a drop in p-ERK1/2, which provided a better prediction of survival compared to the pre-treatment basal p-ERK1/2 level.

In MC9, the level of p-p38 at 24 h was the second most significant feature predicting patient 2y-OS and 5y-OS ($p$ = 3.42e−4, p-adj = 0.0013, Log-HR 3.39 2y-OS and $p$ = 2.969e−4, p-adj = 0.0005, Log-3.50 5y-OS) (Supplementary Fig. 7a–c). In addition, p-Rb ($p$ = 0.003, p-adj = 0.013, Log-HR 1.72) at 24 h in MC9 and the change in Cyclin B1 from pre-treatment to 24 h ($p$ = 0.003, p-adj = 0.014, Log-HR 9.3) in MC8 (NK-cells) were significant in predicting 5y-OS. (Supplementary Fig. 7d, e) ERK1/2 and p38 are both members of the mitogen-activated protein kinase (MAPK) family and play important roles in cell proliferation, differentiation and apoptosis[23]. CREB is a downstream target of multiple kinases, including both p-ERK1/2 and p-p38, and is also directly phosphorylated at S133 by

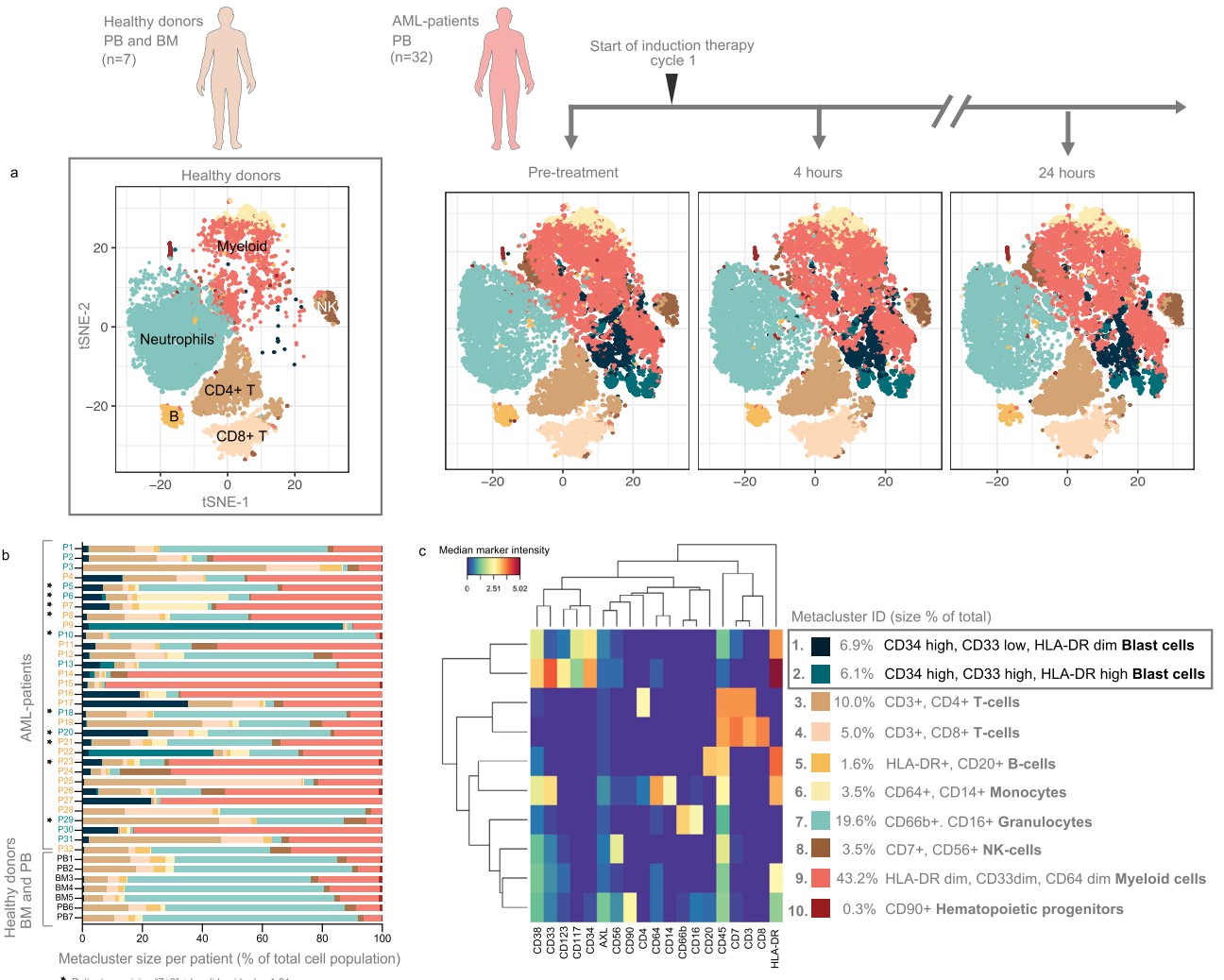

**Fig. 2 | Characterization of immunophenotype in 32 AML patients by FlowSOM. a** t-SNE maps of the 10 FlowSOM identified metaclusters based on surface marker expression across the seven healthy donors and the 32 AML patients at pre-treatment, 4, and 24 h (20,000 cells per plot for visualization). t-SNE maps are annotated with color-coded overlay shown in (**c**). **b** Stacked bar chart showing the metacluster relative distribution in each patient and healthy donor, as a percent of the total population in the pre-treatment sample. Patient numbers are colored by patient 5-year survival (blue = alive, orange = deceased, BM = bone marrow, PB = peripheral blood). The ten patients who received lenalidomide treatment in addition to standard induction chemotherapy ("3 + 7") are annotated by stars. **c** Heatmap showing the median marker intensity of the 19 surface markers used for clustering in the pre-treatment sample of the 32 AML patients. The total metacluster size among the 32 AML patients is shown as a percent of the total. Source data are provided as a Source Data file.

MSK1, MSK2, or MAPKAPK2 (MK2) immediately downstream of both ERK1/2 (MSK1/2) and p38 (MSK1/2 and MK2)[24–26]. Our data confirmed that p-CREB (S133) was significantly higher in the patients with high p-ERK1/2 and p-p38 in MC9 at 24 h (Supplementary Fig. 7f). The ERK pathway has been shown to drive the selection of resistant clones during induction therapy for AML in vitro and elevated ERK1/2 activity has also been observed in AML patients who have developed resistance to FLT3-targeted inhibitors[27,28]. Expression of p38 has also been shown to drive chemotherapy resistance in different cancers[29,30]. Similarly, the high ERK 1/2 and p38 activation levels following chemotherapy treatment could indicate an enhanced pro-survival signaling response in the PB leukemic blasts.

### Immunophenotypic characterization of MC 9

The results of the LASSO Cox regression model indicated that MC9 was a particularly interesting cluster. However, this myeloid cluster was not well defined, and was characterized by a heterogeneous immuno-phenotype across the cohort (Supplementary Fig. 2). Thus, one could question whether the identified p-ERK1/2 association to survival could be attributed to a specific cellular subset within MC9. Therefore, we performed a further in-depth characterization and analysis of MC9 across the patient cohort.

First, we manually gated the pERK1/2 positive and negative cells in MC9 in each patient sample at 24 h (the timepoint where pERK1/2 was significant) to investigate the immunophenotype of these cells. Like the bulk of MC9, we found that both the pERK1/2 negative and positive subset of cells in MC9 had heterogenous immunophenotypes across the cohort (Supplementary Fig. 8). Comparing the surface marker expression levels of the pERK1/2 positive and negative populations (unpaired t-test), we found that the expression of AXL ($p = 0.0002$), CD90 ($p = 4.7E−05$), and CD56 ($p = 0.0006$) was significantly higher in pERK1/2 positive cells. Furthermore, when performing a paired t-test we additionally found that the expression of CD34 ($p = 0.002$) was significantly higher in pERK1/2 positive cells. This suggests that these markers could be associated with suboptimal chemotherapy response, which is supported by previous reports associating all these markers with adverse prognoses in AML.

Next, we set out to further decipher the immunophenotypic diversity within MC9 by dividing this cluster into smaller sub-clusters (Sub-Cs). We performed a new FlowSOM clustering using only the cells

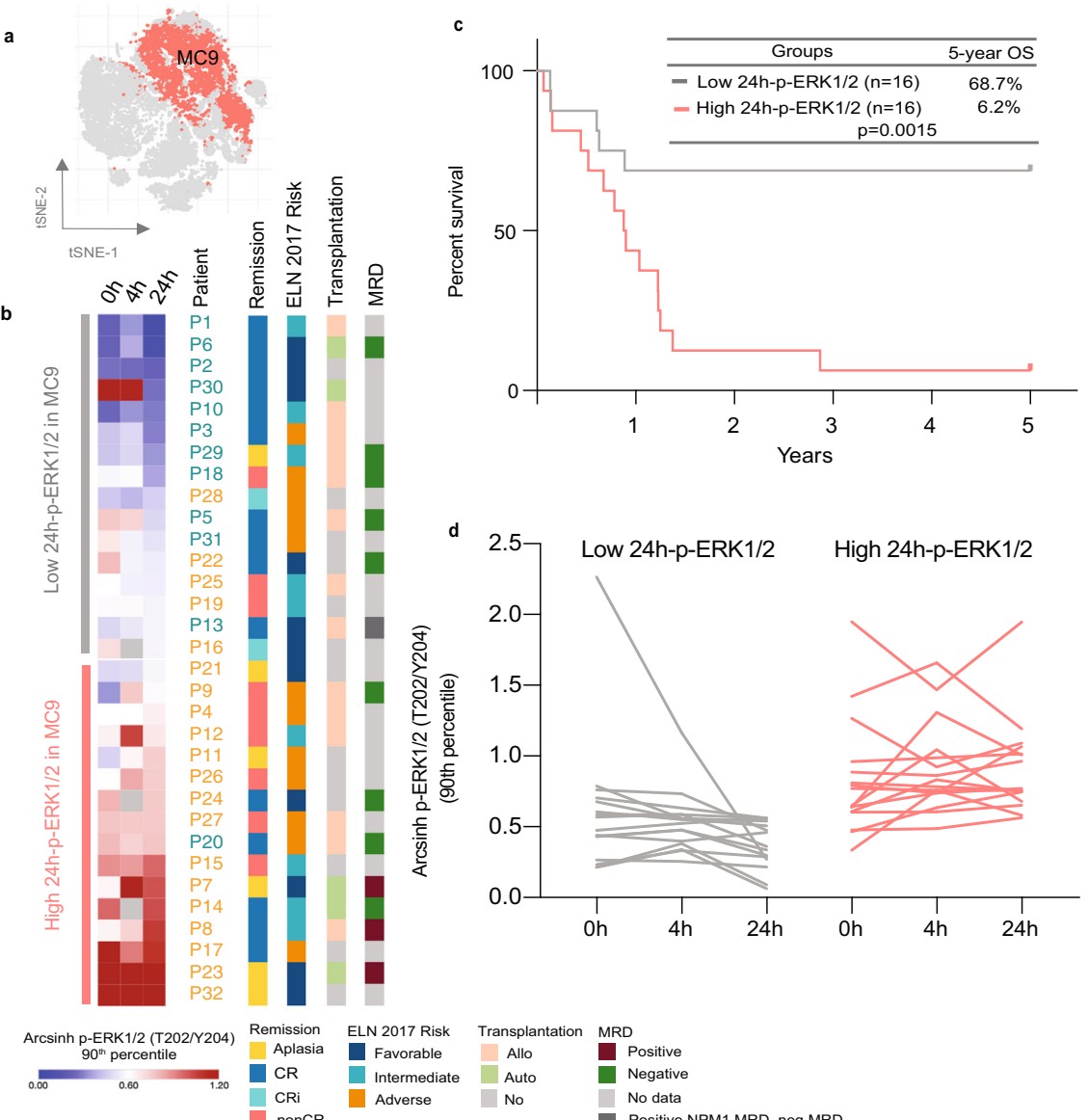

**Fig. 3 | p-ERK1/2 signaling at 24 h after the start of induction therapy predicts patient 2- and 5-year survival. a** t-SNE plot highlighting the position and distribution of metacluster (MC) 9 (red) in the t-SNE plot of all 32 patients at pretreatment. The other metaclusters are shown in gray. **b** Heatmap of the arcsinh transformed 90th percentile p-ERK1/2 in MC9 at all timepoints sorted by the 24 h value, divided by median into high and low 24 h-p-ERK1/2 groups. Patient numbers are color coded by 5-year survival (blue = alive, orange = deceased). Complete remission (CR) by bone marrow aspiration at day 17 or before cycle two of induction therapy, European Leukemia Net (ELN) 2017 risk classification, transplantation status, and minimal residual disease (MRD) status after cycle 2 are shown in color-coded bars to the right. **c** LASSO Cox regression analysis identified p-ERK1/2 at 24 h in MC 9, as significantly associated with patient 2 and 5-year

survival. The 32 AML patients were divided into two groups based on the 24 h median value as shown in (**b**); high and low 24 h-p-ERK1/2 groups, with 16 patients in each group. Kaplan–Meier survival curve of the 5-year overall survival (OS) in the two groups (Log-rank (Mantel–Cox) test *p* = 0.0015, hazard ratio (log-rank) low/high group 0.2307 and 95% CI of ratio 0.0945 to 0.5625) show the poor survival of the high group. **d** Line-graph of the arcsinh transformed 90th percentile p-ERK1/2 value at all timepoints in the two groups. *Abbreviations* Allo allogenous stem cell transplantation, Auto autologous stem cell transplantation, CRi complete remission with incomplete count recovery, nonCR >5% remaining blast in the bone marrow after the first cycle of induction therapy. Source data are provided as a Source Data file.

from MC9 in each patient sample. All surface markers were used for clustering, with 10 sub-Cs as output. As expected, most of the cells (61%) clustered together. This predominant cluster (sub-MC3) was present in all patient samples, although at varying sizes (1.6–88.0%), and was characterized by expression of CD64, CD38, CD33, and HLA-DR (Supplementary Fig. 9a, b). Of note, the algorithm also identified three minor clusters characterized by CD34 expression; sub-MC1 (13.04% of the total, CD123+), sub-MC4 (5.99% of the total, CD7+, and CD123+), and sub-MC2 (4.78% of the total, CD34low, CD117+, and CD123+). The size of these clusters, especially sub-MC1, varied substantially across the patient cohort (Supplementary Fig. 9c). As CD34-

expressing cell populations are known to bear prognostic information in AML[31], we considered whether the pERK1/2 and/or p-p38 signal could originate from these cells. Thus, we performed a new LASSO Cox regression analysis, as described above, using the 10 sub-MCs identified within MC9. This analysis confirmed the prognostic significance of pERK1/2 at 24 h (*p* = 7.3e−5, p-adj = 0.0003, Log-HR = 2.78), but within a small sub-cluster; sub-MC7 (3.24% of total). This cluster was present, but small, in all patient samples, and was characterized by CD20 and CD123 expression. p-p38 was found to be of prognostic significance at 24 h (*p* = 7.8e−6, p-adj = 3.93e−5, Log-HR = 4.3) within a separate small sub-cluster; sub-MC2 (4.78% of total). This cluster was CD34 low and

the only cluster with CD117 expression (Supplementary Fig. 9d). These results do not exclude the possibility of CD34 and/or CD117 expression being a relevant feature of the cells demonstrating suboptimal chemotherapy response. However, extensive inter-patient heterogeneity in the cells responding to chemotherapy rather suggests that the intracellular signaling state is poorly reflected by the immunophenotype. This is analogous to the study by Levine et al., where data-driven bioinformatic approaches identified a progenitor-like signaling phenotype in AML, independent of the immunophenotype, with a significant correlation to prognosis[13].

### Clinical parameters related to the p-ERK1/2 level in MC9

To determine the prognostic impact of standard clinical parameters in the high and low 24 h-p-ERK1/2 group, we investigated the distribution of complete remission (CR/CRi) (before the second cycle of induction therapy), minimum residual disease (MRD) after cycle 2, ELN 2017 risk classification and transplantation status between the two groups (Fig. 3b). More patients achieved CR/CRi after induction therapy in the low 24 h-p-ERK1/2 compared to the high (Fisher exact test, $p = 0.032$). Six out of sixteen patients in high 24 h-p-ERK1/2 and 3/16 patients in the low 24 h-p-ERK1/2 groups had remaining blasts (>6%) at day 17 or before cycle two. However, P18 in low 24 h-p-ERK1/2 had 15% BM blasts on day 17, but when reevaluated on day 26, the number had been reduced to below 5%. P25 in the low group had prolonged aplasia, but 30% blasts on day 29 were measured by flow cytometry. MRD data were available for 14 of the patients in this study. 5/7 patients in the low group had negative MRD after cycle 2, the remaining two had no detectable leukemia-associated immunophenotype at diagnosis and negative MRD but positive NPM1 MRD, respectively. Four out of seven patients in the high group had negative MRD and three patients had positive MRD. The three patients with positive MRD were among the six patients with the highest pERK1/2 values in MC9 at 24 h (Fig.3b). Likely due to the low number of patients in this study, there were no significant differences in the ELN risk groups, MRD or allo-HSCT between patients in low and high pERK1/2 group (Fisher exact test). Ten of the patients included in this study were included in the HOVON 132 trial and received a per-oral addition of lenalidomide from day 1 to 21 during induction therapy. These patients were equally distributed between the high and low 24 h-pERK1/2 groups, with five patients in each group, and we could not detect any signaling responses correlating to the lenalidomide treatment. (Supplementary Fig. 10a, b). A Cox proportional hazard model showed no effect of lenalidomide on patient 5y-survival ($p = 0.16$). Furthermore, there were no significant differences in survival between the patients treated with lenalidomide and the other patients in our study. (Log–Rank (Mantel–Cox) test ($p = 0.15$) and Gehan–Breslow–Wilcoxon test ($p = 0.25$) (Supplementary Fig. 10c).

### Mutations and signaling patterns in MC9

To investigate whether activation of signaling pathways was associated with specific mutations, we associated the signaling in MC9 (Myeloid cells) and MC1 (CD34 + blasts) to next-generation sequencing data (TruSight myeloid panel) and cytogenetics (Supplementary Fig. 11a–f). The two patients with the highest p-ERK1/2 levels in MC9 at 4 and 24 h were the only patients in the cohort with NPM1 mutations and wild-type FLT3. Furthermore, the two patients with the highest level of p-STAT5 in MC9 at all time points had FLT3-ITD mutations. STAT5 is known to be activated in FLT3-mutated AML[32]. There were several examples where patients with the same mutations seemed to have a similar signaling pattern at 4- or 24 h; p-STAT5 at 24 h in patients with IDH1/2 mutations (MC9), p-p38 at 4 h in patients with BCOR or BCORL1 mutations (MC9) and p-NF-kB at 24 h in patients with SRSF2 mutations (MC1) (Supplementary Fig. 11b, d, f). Patients with mutations affecting DNA methylation (DNMT3A, IDH1/2, and TET2) seemed to have a strong induction of p-Histone3 in the CD34 + MC1 from 4 to 24 h

(Supplementary Fig. 11e). Due to the complexity of the mutational landscape of these patients, establishing a significant coupling of mutations to signaling would demand a much larger patient cohort. Nevertheless, we identified signaling patterns that appear to reflect the mutational status of the patients, which could be worth further investigation.

### Signaling in CD34+ CD117+ MC1 AML blast cluster

MC1 had high expression of the stem cell markers CD34 and CD117, a cell population that previously has been suggested of prognostic impact (Supplementary Fig. 2)[31]. Although present in all patients, MC1 had a large variation in cluster size between patients, ranging from 0.04% to 35% (Supplementary Fig. 12a). Ten patients had CD34+ cells in MC9, most of them (7/10) were in the high 24 h-p-ERK1/2 group (Supplementary Fig. 12b) (not significant, Fisher exact test). The MC1 cluster size or the change in cluster size during the first 24 h was not significantly correlated to survival. Furthermore, we did a univariate survival analysis of p-ERK1/2 and p-p38 by dividing the patients into two groups by median at 24 h (16 patients in each group). The difference in survival between the two groups was not significant for p-ERK 1/2, but significant for p-p38 ($p = 0.014$) (Supplementary Fig. 12c). Comparing MC1 with MC9, we found that the level of both p-ERK 1/2 and p-p38 was significantly higher in MC1 than in MC9 (paired $t$-test $p < 0.0001$) (Supplementary Fig. 12d, e). p-ERK1/2 has previously been shown to be upregulated in CD34+ cells compared to CD34− cells[27].

### Validation of FlowSOM automatic cell clustering and supervised LASSO Cox regression analysis by manual gating of mass cytometry data

Our unsupervised cell clustering was followed by an analysis with a machine-learning algorithm that identified treatment-induced alterations in intracellular signaling. We evaluated whether a comparable result could be achieved by simple analysis of manual bi-axial gating of the data. Successful manual analysis may suggest that routine flow cytometry could be used for functional response assessment, e.g., by crude gating of leukemic blasts in bivariate plots using CD45 vs. side scatter (SSC). Mass cytometry data does not provide scatter parameters; therefore, we focused our manual gating on CD45 expression, indicative of hematological cell maturity. Cells were gated by CD45 and CD66b, which allowed the identification of the majority of immature cells (CD45 low/CD66b low) and the exclusion of the more mature cell types (CD45 high), including the granulocytes (CD45 low/CD66b high). The gating strategy is visualized in Fig. 4a. Healthy donor samples included in the respective barcodes were used to guide blast cell gating. After gating, we evaluated the measured dual counts (90th percentile) of p-ERK1/2 and p-p38 in time-to-event analyses, where patients were stratified by median marker expression (above/below median), as described above (Fig. 4b, c). The pre-treatment and the 4 h samples were not predictive of patient survival, while the negative prognostic value of high p-pERK1/2 ($p = 0.0011$) and p-p38 ($p = 0.0013$) at 24 h and the ratio of p-ERK1/2 (24 h/0 h) ($p = 0.0005$) was confirmed (Fig. 4d and Supplementary Fig. 18e). Interestingly, p-ERK1/2 and p-p38 at 24 h and the ratio of p-ERK1/2 was even more significant than in the unsupervised machine learning approach described above. Together, these results show that a simple gating strategy, easily achievable by standard flow cytometric methods, is sufficient to detect chemotherapy-induced changes in PB leukemic blasts that could provide early information on therapy response and therapy outcome.

### Target genes of ERK1/2 and p38 signaling are upregulated after the start of induction therapy

To validate the results obtained by mass cytometry, we examined chemotherapy-induced changes in gene expression profiles by RNA sequencing (RNAseq) in a subset of the patients in our cohort ($n = 14$). A grouped students $t$-test performed on all genes ($n = 50.668$) between

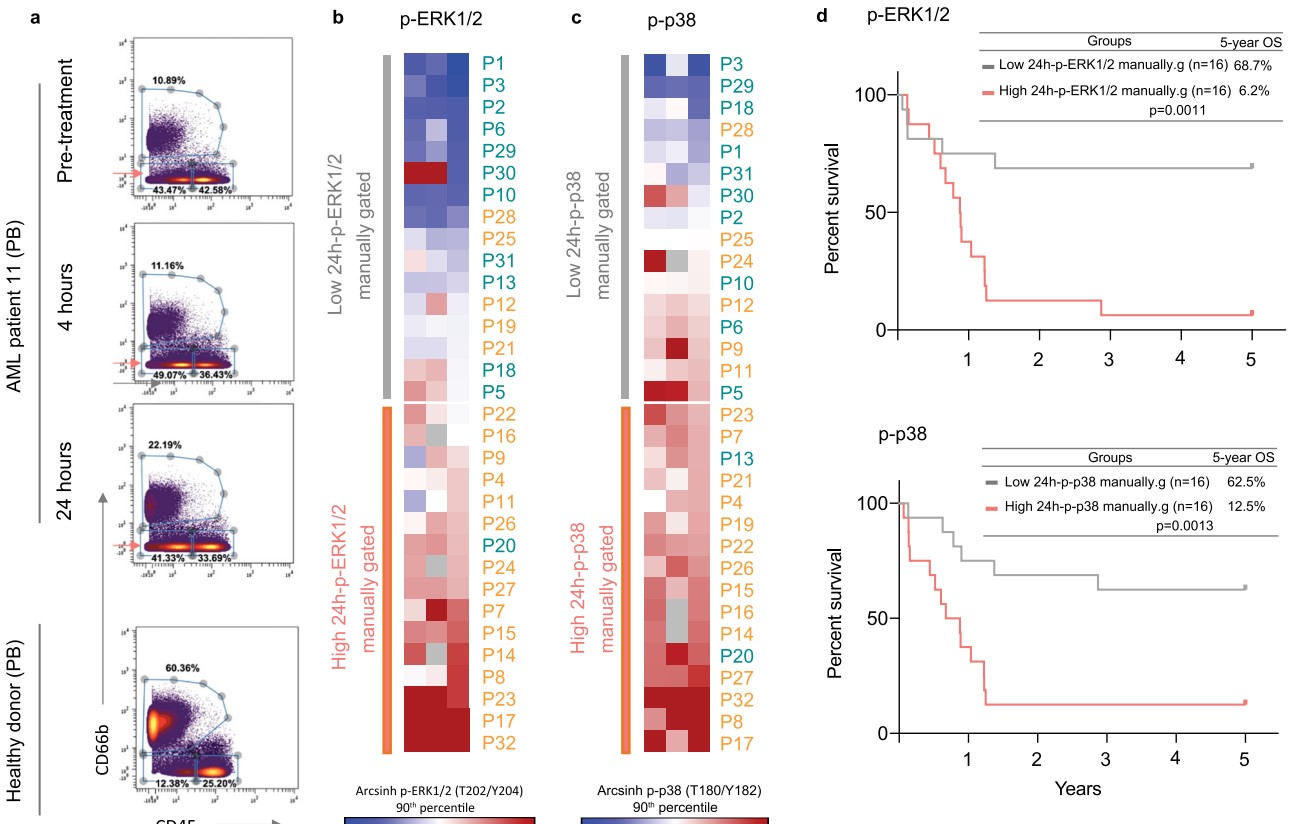

**Fig. 4 | Validation by manual blast gating confirm p-ERK1/2 and p-p38 prediction of survival.** Manual gating of mass cytometry data was performed by bi-axial gating in cytobank on all 32 patients. **a** Bi-axial gating strategy of patient P11 (PB peripheral blood) and a healthy donor (PB) for gating guidance. AML blast cells were defined as CD66 low and CD45 low, the red arrows pinpoint the exported population. Raw 90th percentile data of p-ERK1/2 and p-p38 from the CD66low, CD45low cell population was exported from cytobank and arcsinh transformed with cofactor 5. **b** Heatmap shows the arcsinh transformed 90th percentile data for p-ERK 1/2 and **c** p-p38, sorted by their respective 24 h values. Patient numbers are color-coded by 5-year survival (blue = alive, orange = deceased). **d** Kaplan–Meier survival curves (n = 32) in 24 h-low and high-pERK1/2 group (n = 16) and 24 h-low and high-pp38 group (n = 16). Groups are defined by the median 24 h value for each marker. Curve comparison p-values are calculated by Log–rank (Mantel–Cox) test (p-ERK1/2; HR low/high group: 0.2252, 95% CI of ratio: 0.09215–0.5502, p-p38; HR low/high group: 0.2437, 95% CI of ratio: 0.09791–0.6064). Source data are provided as a Source Data file.

patients in low vs. high 24 h-pERK1/2 groups identified 76 genes with FDR < 0.05. (Supplementary Fig. 13). Among the genes that were high in the high 24 h-p-ERK1/2 groups were HOXA9, which is highly expressed in AML and is known to be a poor prognostic factor[33]. Further on, we focused our analysis on previously reported primary response genes (immediate early genes (IEGs), immediate late genes (ILGs), delayed early genes (DEGs)) and secondary response genes (SRGs) induced by ERK (n = 189) or by p38 (n = 501) signaling[34,35]. In total, this included 689 genes (Supplementary Data 4), of which 525 were identified in our dataset (Fig. 5a). Due to the small sample size (n = 14), the two timepoints post chemotherapy (4 and 24 h) for each patient were used as post-treatment replicates when evaluating chemotherapy-induced changes. A two-sample students t-test was performed on the 24 h-p-ERK1/2 low post-treatment group (n = 3) versus the 24 h-p-ERK1/2 high post-treatment group (n = 7) with a threshold p-value < 0.05 (Fig. 5b). This resulted in 62 significantly differentially expressed genes, of which 29 were upregulated in 24 h-p-ERK1/2 high post-treatment group, including FOSL1 (FOS like 1) (Fig. 5c). FOSL1 is an activator protein-1 (AP-1) transcription factor and an immediate late gene downstream of ERK[34,36]. Patients in the 24 h-p-ERK1/2 high group had a higher induction of FOSL1, especially 4 h post-treatment (Fig. 5d). The analysis also revealed high expression of additional genes that have been linked to poor prognosis in AML in the 24 h-p-ERK1/2 high groups, like heat shock protein 90 alpha family class A (HSP90AA1)[37,38] and F2R like trypsin receptor 1 (F2RL1)[39] Furthermore, we performed a paired students t-test on the 4 h versus pre-

treatment sample in the 24 h-p-ERK1/2 high groups on the 689 ERK- and p38-inducible genes (Supplementary Fig. 14a, b). This analysis revealed another AP-1 transcription factor; activating transcription factor 3 (ATF3)[40], which was strongly induced at 4 h in all patients in the 24 h-p-ERK1/2 high group and decreased in the two patients in the low group that had 4 h samples (Supplementary Fig. 14c, d). Several of these genes, including ATF3, were also upregulated 4 h post induction therapy in our previous work[10].

The MAPK pathway is an important regulator of AP-1 transcriptional activation[41]. We, therefore, investigated the other components of AP-1, which appeared to have a similar pattern as FOSL1 and ATF3; with an increase at 4 h in the high group and a decrease in the low group (Supplementary Fig. 15a–c). Interestingly, Duy et al. showed that induction chemotherapy in AML induced a transient senescence-like state in resilient cells that was capable of initiating AML recurrence[42]. ATF3 has been shown to be critical among the AP-1 transcription factors in reconstructing the accessibility of chromatin to promote senescence[43].

**Super-SILAC proteomics revealed the p38 target MAPKAPK2 (MK2) to be significantly induced 24 h after the start of induction therapy**

Additional validation was performed by proteomics profiling and conducted on 15 patients in our study (Fig. 6a). We quantified 7480 proteins, of which 6363 had a quantitative value in at least two samples (Supplementary Data 5). A paired students t-test between the pre-

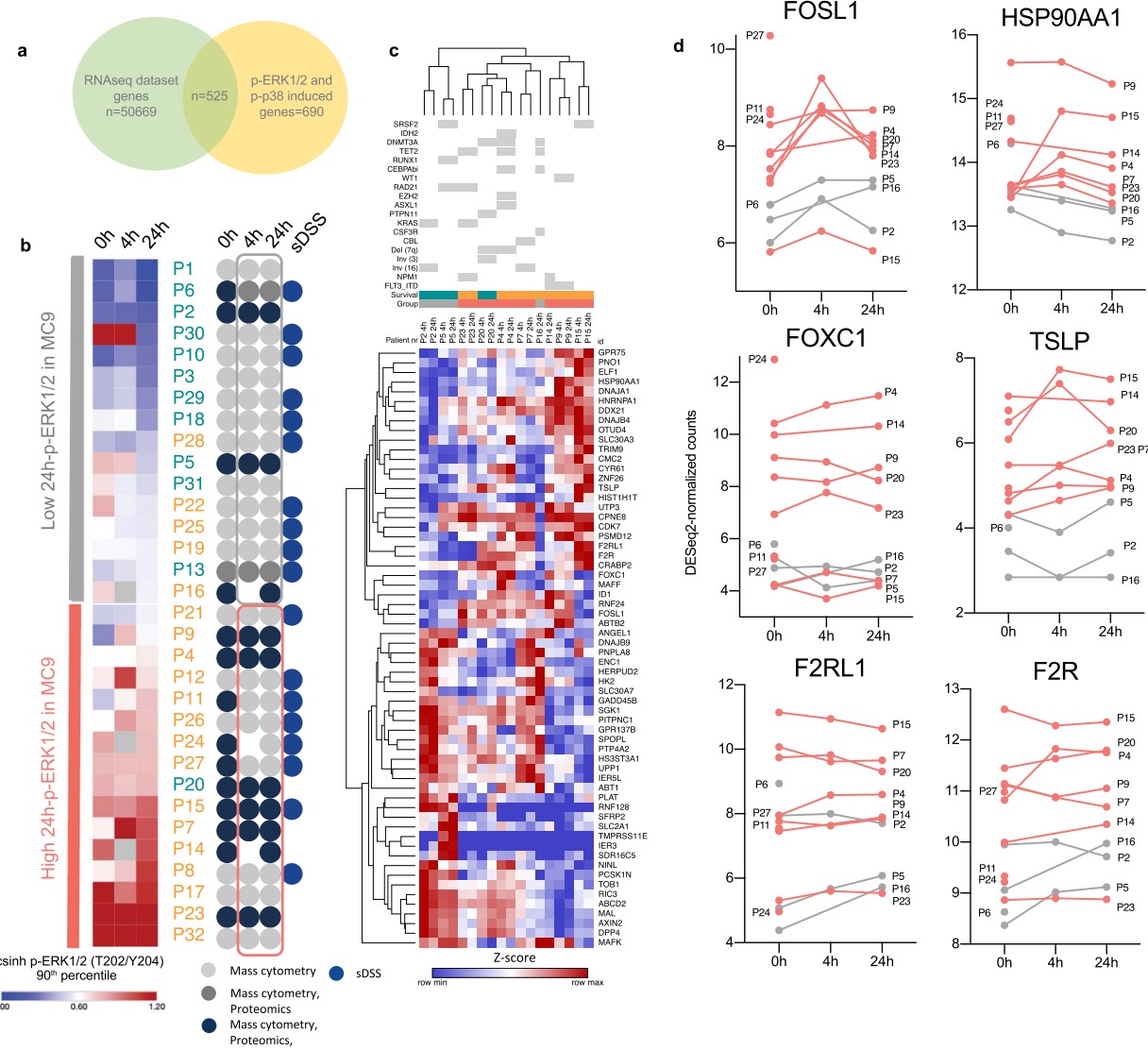

**Fig. 5 | RNA sequencing shows AP-1 complex induction after the start of induction chemotherapy.** **a** Venn diagram showing that 525 p-ERK1/2 and p-p38 induced genes were identified among our RNAseq data. These genes were used for further RNA sequencing data analysis. **b** Heatmap of 90th percentile arcsinh transformed p-ERK1/2 mass cytometry data in M9. The high and low groups used for RNAseq data analysis were defined by the 24 hp-ERK1/2 median value in MC9. Patient numbers are color-coded by 5-year survival (blue = alive, orange = deceased). An overview of the patient sample material and the analysis performed is shown to the right. All 32 patients in the study were analyzed by mass cytometry, 15 patients with mass cytometry and proteomics, and 14 patients with mass cytometry, proteomics, and RNaseq. In addition, diagnostic fresh bone marrow or peripheral blood samples from 18 of the 32 patients was screened with an ex vivo drug sensitivity screen and a selective drug sensitivity score (sDSS). was calculated **c** Hierarchical clustering (Euclidean distance) of the 10 patients with post-treatment samples, based on the gene expression (DESeq normalized counts) of the 62 differentially expressed genes post-treatment identified between the 24 h-p-ERK1/2 high and low groups (only genes with p-values < 0.05 are shown, exact p-values are shown in the source data, two-sided students t-test). The p-values were not adjusted for multiple comparison testing. The heatmap shows z-scored DESeq2 normalized counts. The two timepoints post-chemotherapy (4 and 24 h) for each patient were considered post-treatment replicates. Mutations identified by NGS and diagnostic cytogenetics are shown for each patient in the top panel. **d** Line plots of particularly relevant genes showing the DESeq2 normalized counts at the three different timepoints for each patient. Each line represents one patient, and numbers for identification are annotated next to the lines. For patients with only pre-treatment samples, the patient number is annotated to the left. Patients in the 24 h p-ERK1/2 low group are colored in gray, and patients in the 24 h p-ERK1/2 high group are in red. Source data are provided as a Source Data file.

treatment and 24 h samples in the 24 h-p-ERK1/2 high group (*n* = 7) was performed, with a *p*-value cut off at <0.02. This resulted in 114 differentially expressed proteins; of which 64 proteins increased from pre-treatment to 24 h and 50 proteins decreased (Fig. 6b). MAPK-activated protein kinase 2 (MAPKAPK2), a downstream substrate that is directly phosphorylated and activated by p-38, was one of the most significantly (*p* = 0.0039, p-adj = 1) upregulated proteins at 24 h in the 24 h-p-ERK1/2 high group (Fig. 6b, c). To determine the effect of chemotherapy on cellular proteins, we calculated the change (delta) from pre-treatment to 24 h (24h-0h) for the top 15 most significant proteins. Delta MAPKAPK2, along with ferritin light chain (FTL), had the most significant difference between the 24 h-p-ERK1/2 low groups and 24 h-p-ERK1/2 high groups, with a *p*-value of 0.0091 and 0.0026, respectively (Fig. 6c, d). MAPKAPK2 is a primary target of p38 and has been shown to sustain robust AP-1 activity in triple-negative breast cancer[44,45]. The increase of MAPKAPK2 protein by chemotherapy supports our RNAseq results of induced AP-1 activity post-treatment in the 24 h-pERK1/2 high groups. The p38-MAPKAPK2 signaling axis

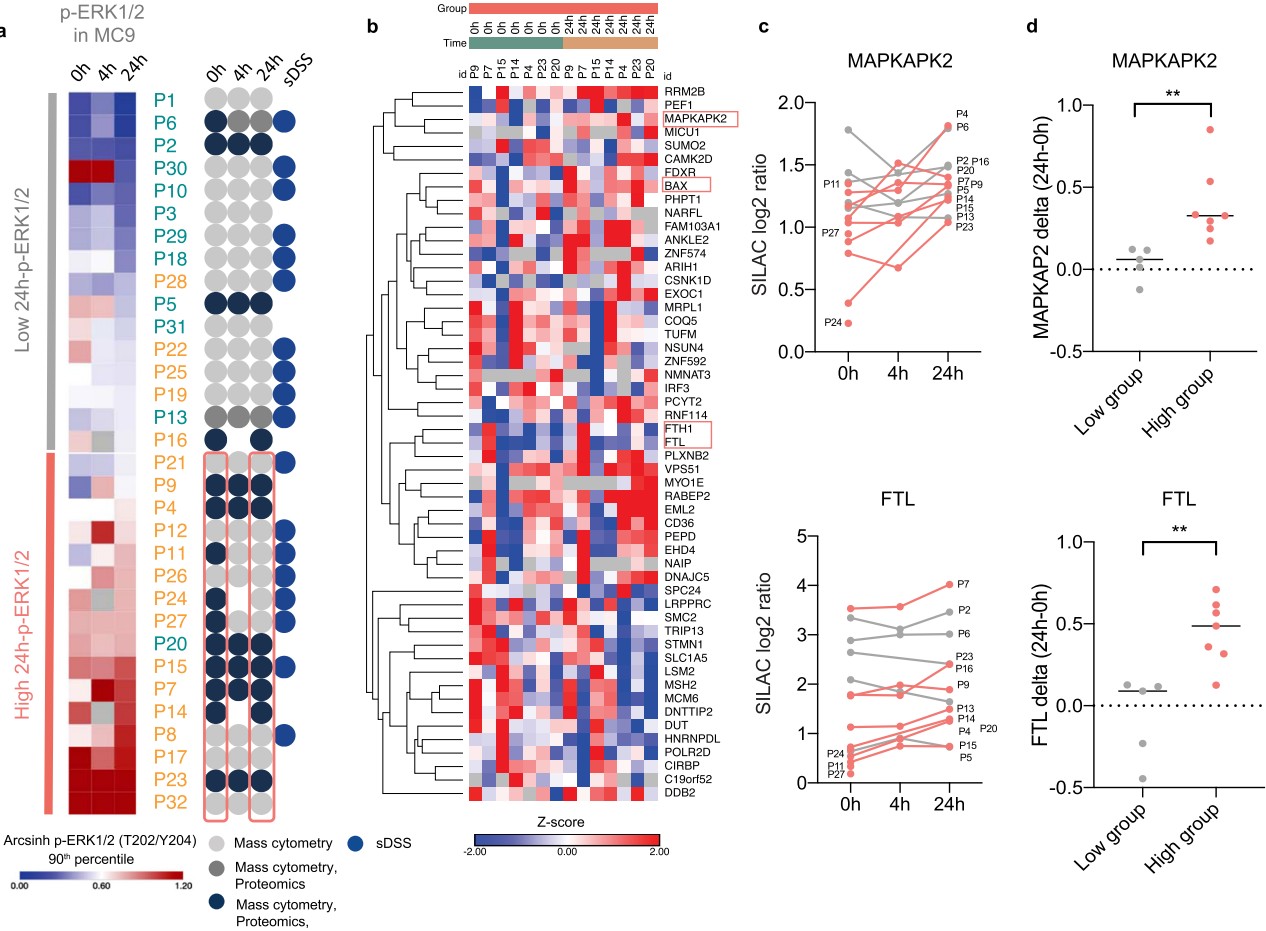

**Fig. 6 | Proteomics reveals increased MAPKAPK2 (MK2) expression at 24 h in patients with high p-ERK1/2 signaling. a** Heatmap of the 90th percentile arcsinh transformed p-ERK1/2 in M9 mass cytometry data. The high and low groups used for proteomics data analysis were defined by the 24-h pERK1/2 median value in MC9. Patient numbers are color-coded by 5-year survival (blue = alive, orange = deceased). An overview of the patient sample material and the analysis performed is shown to the right (sDSS = selective drug sensitivity score). Groups for proteomics paired data analysis are highlighted in red. **b** Hierarchical clustering (Euclidean distance) of the 53 most differentially expressed proteins (Stable isotope labeling using amino acids in cell culture (SILAC) log2 ratio, z-score) identified by a two-sided paired students t-test (only proteins with p-values < 0.01 are shown) between the pre-treatment sample (0 h) and 24-h sample of the seven patients in 24 h p-ERK1/2 high group with 24-h samples. Patients are sorted by sampling

timepoint; 0 h and 24-h. Exact p-values for each protein are shown in the source data. Adjustments for multiple comparisons were not used when identifying these genes. **c, d** The two proteins (MK2 and FTL) among the 15 most significant differentially expressed proteins in the high group (n = 7) that had the most significant increase in expression from pre-treatment to 24 h, compared to the five patients in 24 h p-ERK1/2 low group. **c** SILAC log2 (Light/Heavy) ratio of MK2 and FTL in all patients analyzed by proteomics (n = 15). Patients in the low group are colored in gray and patients in the high group are colored in red. **d** The change in protein expression (SILAC log2 ratio) of MK2 and FTL from pre-treatment to 24 h was calculated and compared between the 24 h p-ERK1/2 low and high groups. Patients in the high group had a significantly higher increase in MK2 and FTL compared to the low group (unpaired t-test, with two-tailed p-value 0.0091 for MK2 and 0.0026 for FTL). Source data are provided as a Source Data file.

has also been shown by others to be induced in quiescent (G0) leukemic cells after exposure to cytarabine, and thereby promote chemoresistance[46]. Interestingly, MAPKAPK2 is especially important in p53-deficient cells for cell-cycle arrest and survival after DNA damage[47]. This is of importance in AML as most patients with TP53 mutations are resistant to intensive chemotherapy as well as allogeneic stem cell transplantation[48,49].

Ferritin heavy chain 1 (FTH1) was the most significantly upregulated protein at 24 h in the 24 h-p-ERK1/2 high group (p = 0.00015, p-adj = 0.98), but did not show the same significance between the low and high 24 h-p-ERK1/2 groups as FTL. Iron is a prerequisite for leukemic cell response to anthracyclines[50], and upregulation of FTH1 may be a protection mechanism that attenuates the effect of chemotherapy. Ferritin heavy/light chain (FTH1/FTL) has previously been linked to cytarabine resistance in AML[51].

The BCL-2 protein family has a central role in the regulation of cell death, and overexpression of BCL-2 in AML is associated with

poor survival and resistance to conventional chemotherapy[52,53]. ERK1/2 and p38 can regulate several members of the BCL-2 family to control cell survival[54,55]. The pro-apoptotic protein BCL-2 associated X protein (BAX) was among the top 15 expressed proteins that increased from pre-treatment to 24 h (p = 0.0022, p-adj=1) in the 24 h-pERK1/2 high group (Fig. 6b). This finding is consistent with our previous work[10]. We, therefore, investigated the other proteins in the BCL-2 family (Supplementary Fig. 16a-d). BCL-2 decreased to undetectable levels at 24 h in 4/5 patient samples in the 24 h-p-ERK1/2 low groups. On the contrary, there were only minor changes in the BCL-2 expression in the 24 h-p-ERK1/2 high group, except in P4 (Supplementary Fig. 16a). Furthermore, expression of the pro-apoptotic BCL-2 antagonist killer 1 (BAK1) was significantly higher from pre-treatment to 24 h in the low 24 h-pERK1/2 groups (p = 0.0145) (Supplementary Fig. 16c, d). These findings reveal that the stress signaling is indeed pro-survival and confers chemoresistance.

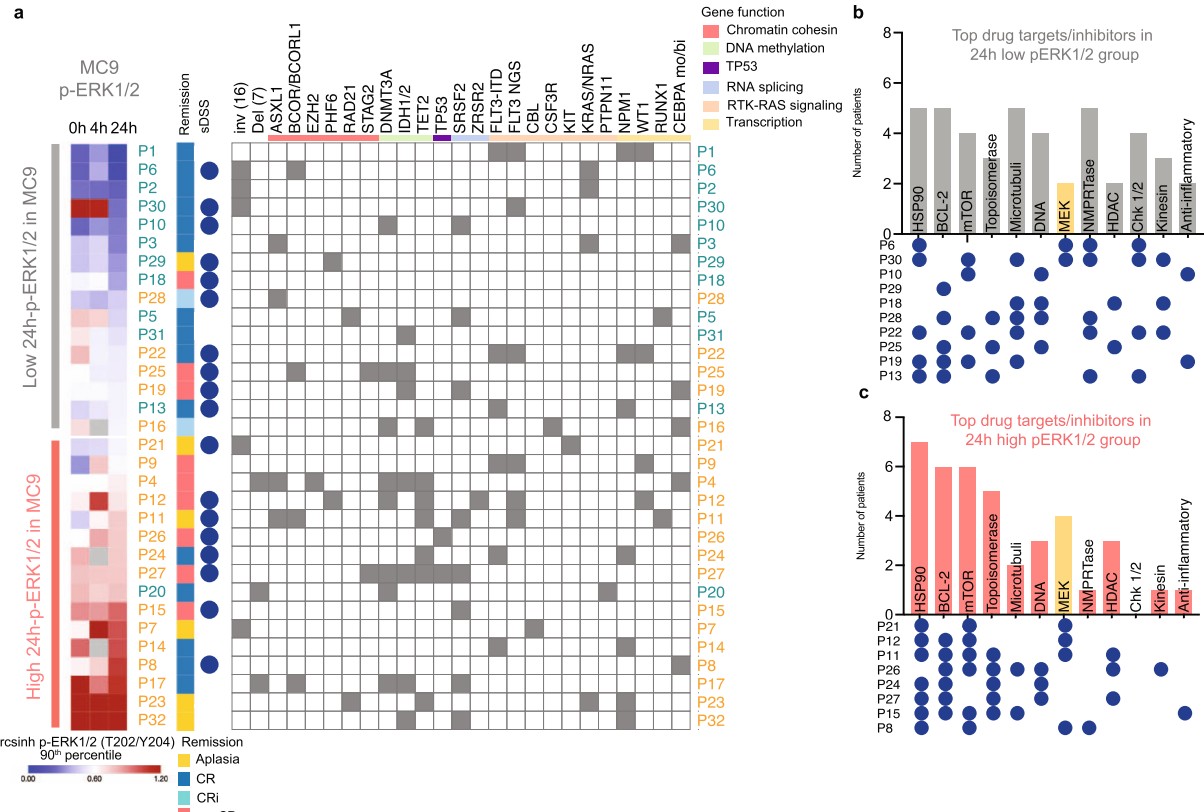

**Fig. 7 | Drug sensitivity and resistance testing (DSRT). a** Heatmap of the 90th percentile arcsinh transformed p-ERK1/2 in M9 mass cytometry data. The high and low groups were defined by the 24-h p-ERK1/2 median value. Patient numbers are color-coded by 5-year survival (blue = alive, orange = deceased). Remission at day 17 post-treatment or before cycle two is annotated to the right. The blue circles indicate the 18 patients that were analyzed by ex vivo drug sensitivity screening. Diagnostic cytogenetics of inv(16), del(7q), FLT3-ITD, and mutations detected by NGS are shown to the right. The column FLT3-ITD annotates the diagnostic cytogenetics, the column FLT3 NGS shows FLT3 mutations detected by NGS (TruSight myeloid panel). Mutations are grouped by gene function, colors are shown in the top panel, and color code is shown to the upper right. Gray squares annotate detected mutations, and white squares if the mutation was not detected. **b,c** The 10 drugs with the highest selective drug sensitivity score (sDSS) for each patient were selected and the 12 most common drug targets in the cohort are shown for the 24h-pERK1/2 low and high group. The circle maps below the x-axis shown which patient had each target among its top 10 drugs. MEK inhibitors are annotated in yellow. **b** Top targets among patients in the low group. **c** Top targets among patients in the high group. *Abbreviations* CR complete remission, CRi complete remission with incomplete count recovery, nonCR = >5% remaining blast in the bone marrow after the first cycle of induction therapy. Source data are provided as a Source Data file.

## Drug sensitivity data show sensitivity for HSP90, mTOR, BCL-2, and MEK inhibitors

Ex vivo drug sensitivity and resistance testing (DSRT) of AML patient samples has been suggested to predict therapy response in vivo[56,57]. We investigated the ex vivo drug sensitivity of a subset of patient samples in the cohort (18/32). Fresh BM or PB from diagnostic samples was screened by a drug sensitivity assay consisting of 349 drugs, using CellTiter-Glo as a readout to assess cell viability. Selective drug sensitivity score (sDSS) was calculated by subtracting the mean DSS of the respective drugs of control samples from five healthy donors (four BM and one PB) from each individual DSS score. (Supplementary Data 6). Notably, we had ten patients with sDSS data in the 24 h-p-ERK1/2 low group and eight patients in the 24 h-p-ERK1/2 high group. sDSS data were correlated to the intracellular signaling profiles detected by mass cytometry by hierarchical clustering (Supplementary Fig 17a, b). As the DSRT was only performed at the time of diagnosis, we chose to compare it with the pERK1/2 value at the time of diagnosis. A grouped students *t*-test between low versus high 0h-pERK1/2 group identified two significant drugs that had a high sDSS in the high 0 h pERK1/2 group, namely the HSP90 inhibitor tanespimycin ($p = 0.0225$, p-adj = 1) and the hypomethylating agent azacytidine ($p = 0.0128$, p-adj = 1) (Supplementary Fig. 17c, d).

Furthermore, we focused on the 10 drugs with the highest sDSS score for each patient (Supplementary Fig. 18). Among the most

frequent compounds with the highest efficacy were HSP90- inhibitors ($n = 12$), BCL-2- inhibitors ($n = 11$), mTOR- inhibitors ($n = 10$), and topoisomerase-inhibitors ($n = 8$). We then investigated if there was a difference in which drugs the patients were more sensitive toward in the 24 h-p-ERK1/2 low and high groups. The most frequent drug targets among patients in the 24 h-p-ERK1/2 low group were HSP90 ($n = 5$), BCL-2 ($n = 5$), and microtubule ($n = 5$) inhibitors (Fig. 7a, b). In the high group, the most frequent targets were HSP90 ($n = 7$), BCL-2 ($n = 6$), and mTOR ($n = 6$) (Fig. 7c). In the high group, 7/8 patients were responsive to HSP90 inhibitors, exemplifying similarities in sensitivity to top targeted drugs. Interestingly, 4/8 (50%) of the patients in the high group had MEK inhibitors among their top 10 targets. Notably, the patient with the highest p-ERK1/2 at 24 h (P8) was very sensitive to MEK inhibitors and had MEK inhibitors among the top three most efficient drugs. For the remaining five patients with the highest p-ERK1/2 level at 24 h measured by mass cytometry, we did not have sDSS data. In the 24 h-p-ERK1/2 low groups, two patients (20%) had MEK inhibitors among their top targets. One of them, P6, had an NRAS mutation and was, as expected, very sensitive to MEK inhibitors. The other patient in the 24 h-p-ERK1/2 low group that had a MEK inhibitor among its top 10 drugs was P30. This patient had the highest pre-treatment p-ERK1/2 value in MC9 of all patients in our cohort. Our data suggest that, in addition to patients with RAS mutations, patients with high p-ERK1/2 levels at diagnosis

or 24 h after induction chemotherapy may benefit from MEK inhibitors.

## Discussion

Conventional cancer response evaluation is based on the measurement of change in tumor load determined clinically, by advanced imaging, cytometry or histopathology, and quantitative genetics[58]. In AML, early evaluation of peripheral blast cell clearance by flow cytometry within the first three days of chemotherapy has been shown to predict CR, relapse risk, and survival[59]. Assessment of tumor clearance is refined in the determination of measurable residual disease (MRD), established as a robust marker for optimal response in leukemia[60] and in solid cancer through the emergence of liquid biopsy and PET imaging[58]. The time it takes to obtain a minimal tumor state may be a challenge for optimal therapy, e.g., after two-month-long cycles of chemotherapy in AML[60], with a risk for tumor regrowth and development of therapy resistance. Measurement of functional cellular responses to therapy may be an alternative and time-saving approach to response evaluation. All cancer therapeutics target cellular proteins directly, from the more selective ATP antagonistic tyrosine kinase inhibitors (TKI) to the multi-targeting antibiotics of anthracyclines. Here, we examined protein phospho-signaling in PB of AML patients before and shortly after the start of therapy, using signaling pathways as sensors for cell fate. Our results show that functional phospho-signaling profiles 24 h post-chemotherapy in this cohort predicted survival more precisely than both risk stratification by BM tumor load on day 17 and the ELN 2017 genetic risk stratification. Thus, in vivo functional cellular responses may provide close to real-time outcome prediction, potentially saving precious time in the evaluation of aggressive tumors.

Biologically relevant intracellular signal systems are affected in vivo early after the start of treatment[11,20,61,62]. In this study, we explored a selection of 15 intracellular targets, covering three major pathways important for myeloid leukemogenesis; the JAK/STAT, RAF/MEK/ERK (MAPK), and PI3K/AKT/mTOR pathways. These signaling systems are tightly connected to the regulation of hematopoiesis and are also affected by mutations in more than 50% of AML cases[9]. AML can be stratified based on signal responses to in vitro stimulation with relevant cytokines and growth factors that regulate these signaling systems, into a potentiated response cluster and a non-potentiated cluster[19]. Analogous lines can be drawn to our study, where the chemotherapy acts as an in vivo stimulus (or stressor) of the signaling networks. We, therefore, hypothesized that temporal dynamics of signaling could be explored in therapy response evaluation. We found that the 24 h cross-sectional analysis resulted in the highest significance in risk stratification. A possible explanation for the higher significance at 24 h may be in the intense cellular stress response caused by the combination of anthracycline and cytarabine. The 24 h chemotherapy treatment may cause an optimal state for decoding the intracellular signaling into precise information about the response. A wide range of cellular stress response systems are activated, and most of the leukemic cell death and removal start beyond 24 h[63]. This is confirmed by our cell profiling, which indicates that the PB cell populations are stable at 4- and 24 h, (Supplementary Fig. 3) and optimal for examination of intracellular signaling perturbations that reflect therapy-induced cell fate.

Functional analyses of cancer cells may improve the genetic diagnostic approach in risk stratification[18]. AML has some of the most robust genetic risk stratification systems in cancer developed over decades, based on a diagnostic sample. However, the prognostic algorithms do not cover a significant proportion of patients with normal cytogenetics and non-detectable mutations by NGS in a satisfactory way. At least 30% of the "good risk" AML patients experience relapse and leukemia-related death[2]. An increasing number of genetic alterations have been included and are incorporated in revisions of the risk stratification. This approach has two possible limitations: First, the analytical process of the stratification will take days to weeks and is most useful for adjustment of consolidation therapy. Second, with higher sensitivity of genetic diagnostics, an increasing number of private genetic aberrations will be discovered: genetic alterations with currently unknown impact in risk assessment. The phenotype of therapy response is defined by a plethora of genetic and epigenetic constitutions. Multivariate single-cell signaling before and early after the start of therapy reflects the genetic and epigenetic script, which we argue defines the therapy response more accurately. To fully utilize the knowledge from genomics, we need to achieve a deeper biological understanding of the connection between cancer phenotype and genotype. Real-time functional response evaluation may be fundamental in deciding the best target for therapeutic intervention among the myriad of genetic lesions[17].

Leukemic cells are characterized by aberrant expression of surface markers, and the intra- and inter-patient immunophenotypic heterogeneity in AML is extensive. This is also evident in the present study. Although presumably healthy cell subsets, such as lymphoid cells in the AML samples, displayed homogeneous expression levels of expected surface markers, malignant cells were characterized by heterogenous surface marker expression patterns and -levels. This reflects that cell identity is not discrete or perpetual but often lies within a continuum[64], such as cells with mixed identities or cells in phenotypic transition. Although expression of isolated surface markers (e.g., CD34, CD117, CD90, CD25, CD7, CD56, and more) has been linked to poor prognosis in AML, the "poor prognostic immunophenotype" in AML remains elusive. This suggests that the link between phenotype and cell functionality is not necessarily straightforward in this disease. Interestingly, previous reports have suggested that a poor prognostic signaling phenotype can be identified in immunophenotypically heterogeneous malignant cell populations in diagnostic AML samples[13] and in response to chemotherapy[17]. Analogously, we identified a functional signaling response with significant prognostic association within MC9, a myeloid cluster with a heterogeneous expression of several markers with the poor prognostic association. Collectively, these results suggest that functional signaling profiles and therapy responses can be identified independently of immunophenotype, potentially providing valuable prognostic information that could supplement conventional outcome assessments.

The possible impact of functional signaling analysis and the prognostic impact of ERK and p38 singling in the prediction of survival needs to be examined in future studies. One limitation of our study is its small cohort size, and follow-up studies are required with larger patient cohorts to further validate our findings. The number of patients included in this study is limited in contrast to the studies that have established genetics in risk stratification and MRD in response evaluation[9,60,65]. This is a possible explanation for the lack of prognostic association with genetics in our cohort. However, the cohort consists of consecutively selected AML patients fit for standard "7 + 3" induction therapy[48], and should, as such, constitute a representative patient cohort. Despite the small sample size, our analyses identified responses with significant prognostic associations. Furthermore, validation by manual clustering, proteomics, and gene expression confirmed the observation of high versus low p-ERK1/2 groups (Figs. 4–6). Thus, we suggest that functional signaling analysis early after the start of therapy could be a future diagnostic tool for improving therapy precision in chemotherapy of AML patients.

## Methods

### Patient samples

The study was performed in accordance with the Declaration of Helsinki, and all samples, including samples from healthy donors, were collected following written informed consent. All patients and healthy donors were above 16 years of age. The biobank and the

clinical protocols were approved by the ethical committee at the University of Bergen (Ethical approval REK Vest 2012/2245, 2012/2247 and the Regional Committee for Medical Research Ethics South-East Norway (REK 2015/2012), the Norwegian Medicines Agency and The Data Inspectorate. Patients were consecutively included in this observational study, and samples were collected between 2014 and 2016 at Haukeland University Hospital, Bergen, Norway, and Oslo University Hospital, Rikshospitalet, Norway. No compensation was given to the patients who contributed to this study. A small fee was given to the healthy donors who volunteered to donate BM samples to the study.

The patients are numbered consecutively after their sampling date. The healthy donors are numbered after the barcode pool (1–7) they were included in. The cohort consist of 32 AML patients (30 de novo and two secondary AML) that received the standard "7 + 3" induction therapy according to the HOVON/SAKK 102 and 132 trials[48], consisting of a 30 min infusion of anthracycline; daunorubicin (60 mg/m$^2$) or idarubicin (10–12 mg/m$^2$) for 3 days in combination with 24 h infusion of cytarabine (Ara-C) for seven days. Routine diagnostic workup of flowcytometry and genetic biomarkers investigated by G-banding, RT-PCR, FISH, and fragment analysis of FLT3 and NPM1 was performed on all patients. 18 of these 32 patients were included in the HOVON 132 clinical trial (EudraCT Number: 2013-002843-26) and were randomized to receive either standard "7 + 3" (Eight patients), or "7 + 3" with the addition of per-oral treatment of lenalidomide at days 1–21 in cycle I and cycle II (ten patients). Lenalidomide treatment was given in the evening on the first day of induction therapy, after the 4 h sample and prior to the 24 h sample. We observed no signaling effect of lenalidomide in our study, and lenalidomide treatment did not have an effect on patient survival. Therefore, we considered patients who received an addition of lenalidomide together with the patients receiving standard "3 + 7". The median age for the 32 AML patients was 56.3 years, there were 19 male and 13 female patients included in our cohort. Supplementary Table 2 shows an overview of the patient characteristics described in Supplementary Table 1 (Supplementary Data 2). Additionally, we analyzed samples from two AML patients who received dose-reduced "3 + 7" (P33 and P34). These were not included in the main FlowSOM and LASSO cox regression analysis but were later added in a new analysis to prove the stability of our findings (described below).

All patient samples in this study were PB samples, collected at the time of diagnosis, and 4 and 24 h after the start of therapy. Three patients (P14, P16, and P24) were only sampled at two timepoints (pre-treatment and 24 h). Furthermore, four PB and three BM samples from healthy volunteers were included in our analyses as healthy reference samples. Additionally, PB from seven healthy donors was used as a control for batch effect standardization.

## Sample preparation for mass cytometry

To minimize potential ex vivo changes in intracellular signaling networks, sample processing was performed within 20 min of sampling. White blood cells were fixed and erythrocytes lysed using Lyse/Fix buffer (BD Phosphoflow), following the manufacturer's instructions. Fixed samples were frozen in saline at −80 °C for shipping and long-term storage. Samples were barcoded using the Cell-ID™ 20 Plex-Pd Barcoding Kit (Fluidigm), following the manufacturer's instructions. The 20 barcoded samples were then pooled, resulting in a total of seven barcodepools. To minimize batch effects, all sequential samples collected from the same patient were contained to the same barcode. Each barcode also included one unique healthy donor sample, either BM (barcodes 3–5) or PB (barcodes 1, 2, 6, and 7). Additionally, an identical reference sample composed of pooled PB from seven healthy donors was aliquoted into all barcodes. This reference sample was used for standardization of batch effects (i.e., staining variability) between the seven barcodepools, as described below.

The samples were stained with an antibody panel consisting of 21 surface markers and 15 intracellular markers (Supplementary table 2). Surface markers known to be expressed by healthy cells, and frequently expressed by AML blast cells were included[66]. The selection of markers was based on EuroFlow panels applied for AML diagnostics[1], and a study performed by Levine et al. that identified markers highly informative in characterizing pediatric AML patients[13]. Samples were stained according to the MaxPar Phospho-Protein Staining Protocol (PRD016 v3) Acquisition of samples was done using a Helios mass cytometer (Fluidigm) at the Flow Cytometry Core Facility, University of Bergen, Norway. All antibodies applied in this study were titrated on lyse/fix PB from five healthy donors, nine of the patients included in this study, and BM from 5\five healthy donors. In addition, all antibodies have been validated by the supplier (please refer to the manufacturer's notes on their website). Several of the antibodies included in this study have been validated by Gullaksen et al. Cytometry part A, 2019. The catalog number and dilutions for each antibody are given in Supplementary Table 3 (Supplementary Data 3). Six antibodies, namely CD14, CD34, pRB, CD7, AXL, and pAxl were conjugated in-house to the isotopes, using the Maxpar X8 antibody conjugation Kit as described by the manufacturer (Fluidigm).

## Mass cytometry data pre-processing and analysis

After the acquisition, the collected data was normalized to EQ bead standard[67]. Cells were gated by DNA-Ir191 versus event length, followed by the exclusion of cell doublets through stringent gating using the two DNA stain channels (Ir191/Ir193). Files were debarcoded using a single-cell debarcode algorithm (https://github.com/nolanlab/single-cell-debarcoder). All data were arcsinh-transformed (cofactor 5). To limit the possibility of spillover between antibodies channels, the data were compensated according to the CATALYST (Cytometry Data analysis tools) pipeline. https://doi.org/10.18129/B9.bioc.CATALYST[68]. The study was performed before the compensation method was available, and we, therefore, used beads created in a later experiment for compensation. The beads were stained with other antibodies conjugated to the same metal isotopes. We assume the difference in metal abundance between the two experiments is negligible, and that the potential difference in metal abundance between the two experiments will be outweighed by the advantage of compensating for the data.

Subsequently, based on the reference samples, all samples were standardized across the seven barcodepools using quantile normalization. Barcodepool 7 was selected as the normalization standard as it had the best separation of populations and the least background staining when the median intensity of all surface markers was compared between barcodes. Standardization was performed following the CytoNorm approach, but without clustering, in 101 quantiles[69]. In short, we calculated 0–100% quantiles for the reference samples in each barcode. For each barcode, we then calculated the piecewise linear function whose value at the $i$th quantile is the $i$th quantile of the reference barcode. Subsequently, the piecewise linear functions were applied to all non-reference samples. The differences in survival outcome based on p-ERK1/2 at 24 h and ratio (24 h/0 h) between unnormalized, uncompensated, and normalized, compensated data are shown in Supplementary Fig. 19. Normalized and unnormalized data without clustering are compared in Supplementary Fig 20a and two tSNE plots showing the 7 reference samples prior to and after normalization is shown in Supplementary Fig 20b. The entire pipeline with clustering was also performed on the normalized, uncompensated data, and the result was p-ERK1/2 at 24 h in MC9 ($p = 0.00133$) with patient 2-year survival.

To identify the immuno-phenotypically defined cell subsets across samples, the single-cell data was clustered using FlowSOM[22]. We used a setup with a 5×5 grid for clustering and consensus meta-clustering with 10 metalusters (MCs). Nineteen of the 21 surface

markers were used for clustering (AXL, CD117 (c-Kit), CD123 (IL-3R), CD14, CD16, CD20, CD3, CD33, CD34, CD38, CD4, CD45, CD56 (NCAM), CD64, CD66b, CD7, CD8a, CD90 (Thy-1), and HLA-DR). CD25 (IL-2R) and CD11b (Mac-1) were excluded as clustering channels due to large variability in staining between barcodes, not correctable by standardization. Clustering and meta-clustering were performed on all cells in the pre-treatment patient samples (baseline) and the healthy donors. We subsequently assigned each cell on post-treatment samples to the closest cluster center. Clustering of data was performed using the FlowSOM package in R (FlowSOM version 2.0.0 https://bioconductor.org/packages/FlowSOM/, R version 4.1.0 (2021-05-18) https://www.r-project.org/)[22]. t-SNE plots showing surface marker expression in these 10 MCs are shown in Supplementary Fig. 21. Supplementary Fig.22 and 23 show color-coded overlay of the seven different batches (barcodepools) and patient ID in these 10 MCs. Supplementary Fig.24 shows a color-coded overlay of the different MC9 sub-clusters shown in Supplementary Fig. 9.

For survival analyses, we applied a Lasso–Cox regression model with automatic feature selection and nested leave-one-out cross-validation to determine the regularization parameter. Since we used leave-one-out cross-validation and only had 32 patients, we could use all possible 1-patient subsets as test samples, all 1-patient subsets as validation samples, and all 30-patient subsets as training data. (Supplementary Fig.5).

Feature input in the model included, for each MC, the measured dual count (arcsinh transformed, 90th percentile) of the intracellular markers cCaspase3, CyclinB1, p-4E-BP1(T37/T46), p-AKT(S473), p-AXL(Y779), p-CREB(S133), p-ERK1/2(T202/Y204), p-Histone3(S28), p-NF-kB p65(S529), p-p38(T180/Y182), p-Rb(S807/S811), p-S6RP(S235/S236), p-STAT1(Y701), p-STAT3(Y750) and p-STAT5(Y684). Additionally, we included the change in the level of each feature from baseline to 24 h (delta 24 h), size of the 10 MCs, and age and sex as features in the model. Survival of the individual patients was followed until five years post induction chemotherapy. For each selected feature, we performed time-to-event analyses in a regularized Lasso Cox proportional hazards model[70]. We split the resulting features that were significantly associated with survival at the median and plotted the survival curves for patients with high and low values of each feature. We used the R package glmnet version 4.1-2. https://glmnet.stanford.edu[71,72].

Our analysis was initially focused on the 32 AML patients who received standard «7 + 3» induction therapy. However, 34 patients were analyzed by mass cytometry in this study. Two of the initially excluded patients, P33 and P34, received a dose-reduced "7 + 3" induction therapy. When these two patients were added to the FlowSOM clustering and we did a new LASSO analysis with these two patients included, we found that p-ERK1/2 (p = 0.0019, p-adj=0.0038, Log-HR 1.25) and p-p38 ($p = 0.0020$, p-adj = 0.004, Log-HR 2.07) in MC 9 at 24 h was significant at predicting 2-year survival. When analyzing 5-year survival with P33 and P34 included, significance was identified in p-ERK1/2 at 24 h in myeloid blasts (MC9) ($p = 0.0011$, p-adj=0.0022, Log-HR 1.29) and p-Rb at 24 h in the B-cells (MC5) ($p = 0.003$, p-adj = 0.006, Log-HR 1.67).

Furthermore, we performed a new FlowSOM analysis with only the cells from MC9 in all patients at all timepoints. We used the same parameters for clustering as described in the first analysis above. T-SNE plots showing the distribution of patients in each of the seven barcodepools (batches) are shown in Supplementary Fig. 25. The staining of pERK1/2 in MC9 for all patients at all timepoints and healthy donors is shown in Supplementary Fig. 23.

## Drug sensitivity and resistance testing (DSRT)

Cells derived from 18 of the patients included in our study were screened for sensitivity to 349 drugs (anti-cancer compound library #L3000 from Selleck Chemicals) using DSRT[16].

The 349 compounds from the anti-cancer Selleck Chemical library were pre-aliquoted at five concentrations, increasing in 10-fold steps from 1 nM to 10 μM, in 384-well plates using an Echo Liquid Handler. Each plate also contained eight positives (benzethonium chloride, BzCl) and eight negative (DMSO only) controls. Fresh diagnostic (pre-treatment) BM or PB was lymphoprepped (15 BM and 8 PB), and cells were seeded directly into the plates containing the drugs at a concentration of 10.000 cells/well and incubated for 72 h in mononuclear cell media (MCM) media (promo cell C-28030) supplemented with 1% Penicillin + Streptomycin (PS) (Gibco,15140-122). Finally, cell viability was assessed by CellTiter-Glo Luminescent viability assay. All drugs were also tested against samples from five healthy donors (four BM and one PB).

All patient samples were normalized to the median of the positive (BzCl) and negative (DMSO) controls per plate and presented as selective Drug Sensitivity Score (sDSS), calculated using the online tool Breeze (https://breeze.fimm.fi/28489_mc43oty0nzywmcaxnje3nzgwnzaw/index.php#).

## Patient sample preparation for RNA sequencing, proteomics, and next-generation sequencing (NGS)

PB samples at pre-treatment, 4 and 24 h after the start of treatment for proteomics, RNA sequencing, and targeted DNA-seq were sampled in BC Vacutainer CPT™ Mononuclear cell preparation tubes with sodium heparin. Mononuclear cells were prepared by density gradient separation (Lymphoprep, Axis-shield). For RNA sequencing, approximately $1 \times 10^6$ cells were dissolved in TRIzol™ Reagent (Thermo Fisher) and stored at −80 °C until analysis. For proteomics, approximately $5 \times 10^6$ cells were precipitated in 7% trichloroacetic acid (TCA) and stored at −80 °C until analysis. For targeted DNA-seq, approximately $5 \times 10^6$ cells were pelleted, the supernatant removed, and stored as a dry pellet at −80 °C until analysis. The gradient separation and sample preparation for storage were done immediately after sampling and the same standardized method for sample preparation was used at both collection sites in Oslo and Bergen.

## TruSight myeloid panel sequencing

Totally, 54 frequently mutated genes in myeloid malignancies were sequenced using Illumina Trusight Myeloid Gene Panel with Miseq where v3 reagent kit was used (all from Illumina, San Diego, CA, USA)[73]. The amplicon sequencing library was prepared from 50 ng DNA according to the manufacturer's instructions, except for normalization, which was done manually by measuring each sample on Qubit and thereafter diluting each sample to a concentration of 4 nM. Sixteen samples were sequenced each time and the total DNA input on the flow cell was 15 picomolar. Secondary analysis was done using MiSeqReporter version 2.6.2.3 (Illumina) mapping to the human genome reference hg19 and variant calling using Somatic variant caller 3.5.2.1. Sequence alignment of selected variants was manually examined with the Integrative Genomics Viewer (IGV)[74]. Annotation was done by snpSIFT and snpEFF v 4.1. Variant filtering and interpretation were done in Coremine Oncology (PubGene AS, Oslo, Norway) with >1% minor allele frequency in the 1000 genomes data were presumed to be germline and removed from further interpretations. Synonymous substitutions and intronic variants that were not at the splice site and variants that were interpreted either as benign or most likely benign were not included. The variant allele frequency (VAF) was calculated for each mutation as the number of variant reads divided by the total reads. The cut-off for reported variants was for VAF >8% with a reading depth of 100. Only variants interpreted as pathogenic, probably pathogenic, and variants of unknown significance were reported. The nomenclature is according to HGSV (http://varnomen.hgvs.org). FLT3 and NPM1 were also investigated by fragment analysis and CEBPA by sanger sequencing.

## Super-SILAC proteomics sample preparation and data analysis

TCA-precipitated samples for proteomics were washed three times in cold water-saturated ether prior to proteomics sample preparation[20]. Samples were then lysed in 4% sodium dodecyl sulfate (SDS)/0.1 M Tris-HCl (pH 7.6), heated at 95 °C for 7 min under mild shaking and sonicated (3 cycles at 30% amplitude for 30 s with 1 min rest between each cycle). Cell debris was removed by centrifugation (14.000×*g* for 10 min), and the protein concentration was determined with the Pierce BCA Protein Assay kit (Thermo Fisher Scientific). Samples were kept at −80 °C until analysis. For labeled proteomics, 20 mg of each patient sample was mixed with 10 mg of a super-SILAC mix composed of five heterogenous AML cell lines labeled with Arg6 and Lys8 isotopes. The patient cell lysate was further processed by filter-aided sample preparation (FASP) and small-scale proteome fractionation[75].

Abundance and peptide identification was conducted on a Q Exactive HF Orbitrap mass spectrometer coupled to an Ultimate 3000 Rapid Separation LC system (Thermo scientific)[75].

LC–MS/MS raw files were processed with MaxQuant software version 1.5.2.8[76]. The spectra were searched against the concatenated forward and reversed-decoy Swiss-Prot Homo sapiens database (version 2018_02), using the Andromeda search engine[76]. The Perseus 1.6.1.1 platform was used for data analysis and statistics. MaxQuant-normalized SILAC ratios were inverted, log2 transformed, and normalized again, using width adjustment.

## RNAseq data analysis

36 samples were sequenced in Hiseq 4000 with 75 × 2 chemistry. The demultiplexed fastq files were aligned to the human genome GRCh38.p13 using HISAT2 aligner[77]. The aligned bam files were quantified as a whole read count matrix using the tool FeatureCount[78]. The generated read count was normalized using DESEQ2 R package. Variance stabilizing transformation was adopted. *P*-values below 0.05 were considered significant. The Perseus 1.6.1.1 platform was used for RNAseq data analysis and statistics.

## Reporting summary

Further information on research design is available in the Nature Portfolio Reporting Summary linked to this article.

## Data availability

The mass cytometry data generated in this study have been deposited in the FlowRepository database under accession code RvFr0LLv9McDJ89jgK50G4lwnfDFRTrcMelxYgnSIcE2Cymrpf2qh2NaWybtWDNH. The proteomics data generated in this study have been deposited in the ProteomeXchange Consortium via the PRIDE partner database under accession code PXD031916 Proteomics output data is provided in this manuscript as Supplementary information file 1. The raw fastq files from the RNA sequencing data are deposited in the NCBI gene expression omnibus (GEO) repository and is available under the accession code GSE218664 The DSRT data and the results of the NGS (Illumina TruSight myeloid panel) data generated in this study are provided in the Source Data file. Public availability of raw data DNA sequencing data (TruSigh myeloid panel) is not compliant with Norwegian regulations (GDPR) or allowed by the patient consent. For non-commercial academic use, please contact B.T.G. for further information (bjorn.gjertsen@uib.no), which will require an ethical application to the Regional Committee for Medical Research Ethics in Norway (REK). The time the data will be available for the requester will need to be provided in the application and discussed with REK, and further information can be found on the REK website [https://rekportalen.no]. In the proteomics data analysis, the spectra were searched against the concatenated forward and reversed-decoy Swiss-Prot Homo sapiens database (version 2018_02), using the Andromeda search engine. The RNA sequencing fastq files were aligned to the human genome GRCh38.p13 using HISAT2 aligner. Secondary analysis of the TruSight myeloid data was done using

MiSeqReporter version 2.6.2.3 (Illumina) mapping to the human genome reference hg19 and variant calling using Somatic variant caller 3.5.2.1. The remaining data are available within the Article, Supplementary Information or Source Data file. Source data are provided in this paper.

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

## Acknowledgements

This work was supported by grants provided by EU ERA PerMed AML_PM (RCN Grant no. 298842), the Research Council of Norway (grant no. 303358, 294916), the Norwegian Cancer Society with Solveig & Ole Lunds Legacy, Øyvinn Mølbach-Petersens Fund for Clinical Research (grant No. 303445 and 190175-2017, 182524, and 208012), Western Norway Health Authorities (grant no. 912308 and F-12148). Norwegian Health Authority South-East, (grant no. 2018012 and 2019096). This work was partly supported by the Research Council of Norway through its Centres of the Excellence funding scheme, (project number 262652). The authors thank nurses and physicians in Bergen and Oslo, personnel at the Flow Cytometry Core Facility at the University of Bergen, and personnel at the Gjertsen- and Enserink labs for their skillful technical assistance. The authors would also like to acknowledge Clinical Genomics Uppsala, the Science for Life Laboratory, Department of Immunology, Genetics and Pathology, Uppsala University, Sweden, for performing NGS and The Genomics Core Facility (GCF) at the University of Bergen, which is a part of the NorSeq consortium, provided services on RNAseq; GCF is supported in part by major grants from the Research Council of Norway (grant no. 245979/F50) Bergen Research Foundation (BFS) (grant no. BFS2017TMT04 and BFS2017TMT08). The authors thank the Norwegian blood cancer association, the University of Bergen, and the medical student research program at the faculty of medicine. Thanks to Dr. Jacqueline Cloos, Rotterdam, and HOVON for the MRD data.

## Author contributions

The study was designed by B.S.T. and B.T.G., and the paper was composed primarily of B.S.T., M.H., and B.T.G. Patient samples were collected and processed by B.S.T. and O.H.E.F. in Bergen and P.A.D., D.S.T., and L.P. in Oslo. B.S.T., M.H., O.H.E.F., J.S., and S.E.G. performed mass cytometry experiments. N.B. wrote R scripts for mass cytometry data analyses. B.S.T., M.H., and B.T.G. analyzed mass cytometry data. A.S. and B.S.T. performed an analysis of RNA sequencing data. BST performed proteomics wet lab, and E.B. and B.S.T. analyzed data. P.A.D., L.P., and D.S.T. performed drug sensitivity and resistance testing, which was supervised by J.M.E. R.H. and P.B. contributed to molecular diagnostics and analyses of sequencing data. D.K. contributed to data analysis of sequencing data. Y.F. was responsible for the collection of biological samples and patient treatment in Oslo. T.H.A.T. collected clinical data from patients treated in Oslo. V.A., O.M.S., N.A., S.G., K.P., I.J., and J.E. contributed to data interpretation and writing. All authors contributed to the paper preparation.

## Funding

## Competing interests

The authors declare no competing interests.

## Additional information

[1]Centre for Cancer Biomarkers (CCBIO), Department of Clinical Science, University of Bergen, Bergen, Norway. [2]Department of Internal Medicine, Hematology Section, Haukeland University Hospital, Helse Bergen HF, Bergen, Norway. [3]Genome Core Facility, Clinical Laboratory, K2 Haukeland University Hospital, Bergen, Norway. [4]The Proteomics Facility of the University of Bergen (PROBE), University of Bergen, Bergen, Norway. [5]Centre for Cancer Biomarkers and Computational Biology Unit, Department of Informatics, University of Bergen, Bergen, Norway. [6]Department of Molecular Cell Biology, Institute for Cancer Research, The Norwegian Radium Hospital, Montebello 0379 Oslo, Norway. [7]Centre for Cancer Cell Reprogramming, Institute of Clinical Medicine, Faculty of Medicine, University of Oslo, 0318 Oslo, Norway. [8]Faculty of Medicine, University of Oslo, Oslo, Norway. [9]Department of Molecular Genetics, Division of Laboratory Medicine, Oslo University Hospital, Oslo, Norway. [10]Department of Translational Hematology and Oncology Research, Cleveland Clinic, OH 44106, USA. [11]Department of Immunology, Genetics and Pathology, Science for Life Laboratory, Uppsala University, Uppsala, Sweden. [12]Center for Medical Genetics and Molecular Medicine, Haukeland University Hospital, Bergen, Norway. [13]Department of Pharmacy, UiT-The Arctic University of Norway, 9037 Tromsø, Norway. [14]Department of Hematology, Oslo University Hospital, Oslo, Norway. [15]Department of Anesthesiology, Perioperative and Pain Medicine, Stanford University School of Medicine, Stanford, CA 94121, USA. [16]Department of Pediatrics, Stanford University School of Medicine, Stanford, CA 94121, USA. [17]Department of Biomedical Informatics, Stanford University School of Medicine, Stanford, CA 94121, USA. [18]Centre for Clinical Treatment Research (NeuroSysMed), Department of Clinical Science, University of Bergen, Bergen, Norway. [19]Department of Hematology, Helsinki University Hospital Comprehensive Cancer Center, Helsinki, Finland. [20]Section for Biochemistry and Molecular Biology, Faculty of Mathematics and Natural Sciences, University of Oslo, 0037 Oslo, Norway. [21]Department of Informatics, University of Bergen, Bergen, Norway.
✉e-mail: nello.blaser@uib.no; bjorn.gjertsen@uib.no

