## [Peer Review File · Nature Communications]

Early response evaluation by single cell signaling profiling in acute myeloid leukemiaReviewers' Comments:

Reviewer #1:

Remarks to the Author:

In this manuscript, the authors used mass cytometry to measure the abundance of 36 extracellular and intracellular proteins in the peripheral blood of 32 AML patients at 0, 4, and 24 hours after treatment with 7+3 chemotherapy. Using LASSO Cox regression, p-ERK1/2 and p-p38 abundances in one myeloid cell population (called here "MC9") at one time point (24 hr) were found to be significantly correlated with survival. This paper is well written and clear. The overall result is statistically significant and interesting, and suggests that this proteomic approach may become very important to the field. That result does not appear to be extremely robust given the arbitrary nature of the clustering analysis, and the lack of correlation with chemotherapy response, ELN risk status, transplantation status, or mutational status. As the authors note, this may simply be due to the size of the cohort, which is small relative to the heterogeneity and complexity of this disease. The authors also demonstrated that p-ERK1/2 and p-p38 abundance in flow-sorted blasts is correlated with survival, which strengthens the conclusions.

Specific comments:

- MC9 is surprisingly heterogeneous with respect to CD34 expression. Because clustering is arbitrary, and clusters do not represent an absolute "ground truth," this heterogeneity warrants further investigation – for instance, given the heterogeneity of this important marker, can the authors justify calling this one cluster? In general, MC9 should be better characterized, for example, by highlighting each patient on the tSNE plot, and by showing the expression of each analyte, particularly CD34 and p-ERK1/2, on the tSNE plot, so that we can better understand heterogeneity within that cluster.
- What proportion of malignant cells does MC9 represent in each patient? Does this proportion correlate with survival? Does change in the size of MC9 over time relate to survival? Does this depend on the extent of CD34 expression within MC9? One limitation of the proteomic approach is that it is difficult to know exactly which cells are malignant, because no genetic/mutational information is available, although sorting and targeted sequencing for somatic mutations could be performed to address this further.
- Further discussion of the specific relevance of MC9 to survival would be helpful. For example, why is protein expression in this cluster more relevant to survival than, say, the size or characteristics of the blast cluster, MC1? Is change in the relative size of MC1 and MC9 (over time) correlated with survival?
- High and low p-ERK1/2 patients were defined relative to each other, depending on whether they were above or below the median in this cohort. Is there an absolute protein abundance that can be used to determine whether new patients are "high" or "low?" If not, how would new patients be classified with respect to p-ERK1/2 abundance? The result would be stronger if there were an additional validation cohort that could be independently classified with respect to p-ERK1/2 and p-p38 abundance.
- Mutation correlations were very anecdotal. Obviously, this analysis would benefit from more patients and exome or WGS sequencing.
- The confirmatory flow cytometry analysis on pp. 14-15 actually gave a more significant result than the original mass cytometry clustering analysis. This strengthens the overall claim of the paper, but raises some interesting questions. How do the immature cells defined by this blast gate correspond to the FlowSOM clusters shown earlier? Are these flow-gated blasts in MC9 or MC1? The authors could reanalyze the cytometry data in a manner analogous to the flow sorting – this might strengthen the result, or would at least add clarity by connecting these two analyses to each other.
- For the bulk RNA-seq analysis, a more unbiased analysis could be useful. What other genes were differentially expressed between the p-ERK1/2 low and high patients? Moreover, if the samples are hierarchically clustered based on the bulk RNA-seq data alone, do they naturally cluster into the same groups (p-ERK1/2 low and high)?
- The bulk RNA-seq t-test comparison essentially assumes that MC9 is roughly the same size in all patients. A bulk analysis that takes into account the size of MC9 in each patient might be more sensitive at detecting genes of interest.

- The drug sensitivity matrix should also be hierarchically clustered by patient and compared to the proteomic data to discover other potential correlates of drug sensitivity in an unbiased fashion.
- The authors repeatedly mention that their proteomic data indicates the presence of “clonal heterogeneity,” but this is traditionally defined by genetics, not proteomics. For instance, clustering of somatic mutations by variant allele frequency would reveal clonal heterogeneity.

Reviewer #2:

Remarks to the Author:

The authors used 36-dimensional mass cytometry data to identify genes sensitive to the chemotherapy response. These were further validated using RNAseq and mass spec. The reason of this study is well explained and further validated using DSRT.

1. The tSNE plot in Figure 1 can be colored with ID to see if clustering is not influenced by batch effect.
2. Many clusters including MC1,2,3 are scattered. 1) Isn't it because of batch effect? coloring based on patient ID will answer. 2) Why they are scattered? Is it due to the assignment to the existing clusters? How then they can be assigned to the same cluster? For instance, MC1 is scattered into to clusters? How these are defined as MC1?
3. MC9 has been extensively used for the analysis. Why MC9 (the myeloid cluster) could become most informative cluster among other MCs? Any speculation on it?
4. Besides 24h-p-ERK1/2, p-p38 at 24, p-Rb at 24h were reported. The combination of several markers could lead a better performance. This has to be tested and discussed. In the same way, can the information obtained from RNAseq and Mass spec be used to in designing combination of markers?

Reviewer #3:

Remarks to the Author:

In the present manuscript the authors assessed the correlation between single cell signaling and clinical response to induction chemotherapy in 32 patients with AML. Samples were collected at diagnosis and during the course of induction chemotherapy, and analyzed by mass cytometry, RNAseq and mass spectrometry proteomics in order to assess 1/ the chemotherapy-induced phospho-signaling modulations in the myeloid compartment and 2/ the prognostic significance of phospho-protein expression levels in myeloid cells 24h after treatment initiation. Overall, the manuscript is well written and the mass cytometry data are presented in a clear way. However some issues detailed below prevent publication of this manuscript in its present form.

Graphical abstract :

- 1/ The legend is constructed as a standard legend; this legend should rather focus on the summary of the key points and the main conclusions of the article.
- 2/ The risk stratification by bone marrow tumor load is mentioned in the discussion section, but has not been displayed in the results section or in the supplementary data. The overall survival according to the ELN classification only appears in the graphical abstract as well, and should be integrated to the body of the article.

Results section

- 1/ Identification of an independent prognostic factor in AML should be confirmed using a Cox

regression including the validated prognostic factors in order to exclude potential confounding factors. In the present work, this Cox regression should include 24h-pERK1/2, age, ELN, WBC count at diagnosis, as well as allogeneic stem cell transplantation, ideally as a time-dependent covariate. Of note, the poor survival in the ELN favorable group, especially taking into account the fact that most patients are young patients, underlies that there might be a bias in the present cohort and probably precludes proper Cox regression for a formal identification of a potential prognostic factor.

2/ The MRD status is a cornerstone for prognostication of AML. An analysis with respect to the MDR would therefore be essential before concluding about the potential usefulness of measuring pERK1/2 to refine the therapeutic decision algorithm in AML.

3/ The authors claim in the abstract that they questioned whether the signaling response to therapy was more informative than analysis at time of diagnosis.

In fig 2d, the basal level of pERK1/2 seems low in the low-24h-pERK1/2 group and high in the high-24h-pERK1/2 group. Did the authors assess the impact of baseline expression in the metacluster 9 on clinical outcome for a direct comparison? This information is important in order to appreciate the improvement of the quality of prediction using samples drawn during induction chemotherapy rather than baseline samples.

4/ Why did the authors choose to include patients with MDS and B-ALL in this manuscript? At least the authors should retain only the 32 patients included in the study in the patient characteristics table.

5/ In supplementary fig 3, the cluster of NK cells is located at 2 different places on the tSNE plot. Some cells that were annotated as NK cells clusterize in the region of myeloid cells, and might be leukemic blasts that express CD56. This would also explain the low CD45 mean expression of this cluster compared with CD45 expression in the clusters of CD4 and CD8 T cells. The authors should confirm by manual gating the exact nature of these cells.

The authors should provide expression of the different markers projected on the tSNE plots in supplementary 3 in order to enable appreciation of the homogeneity of the different clusters.

Minor: in supplementary fig 3, the lack of heatmap annotation in the lower panel makes the figure difficult to read.

6/ In Fig 3, the authors detail the manual gating strategy of leukemic blasts in order to confirm the results obtained with machine learning algorithms and assess the transferability of the results in a routine setting. However, the metacluster used to generate the initial survival curves according to pERK1/2 expression was MC9, which was considered as a myeloid cluster, by contrast with the MC1 and MC2 that were considered as the AML blast clusters (line 198). Could the authors clarify this discrepancy in the selection of the cell population they chose for manual confirmation of the results?

7/ In fig 6, results are presented for 16 and 9 drugs in the low 24h – p-ERK1/2 group and the high 24h – p-ERK1/2 group, respectively. The lack of consistence between panel b and c makes it difficult to compare the graphs. In addition, the drugs that are displayed in figure 6 should be listed in the materials and methods section (“MEK inhibitor”, “DNA inhibitor” or “anti-oxidant” being too evasive). The barplots should be annotated so as the drug class appears as “Protein X inhibitor” rather than “Protein X”.

The aim of this figure is to correlate the intracellular signaling profiles detected by mass cytometry with ex vivo drug sensitivity (line 573). The way the results are presented does not enable a direct comparison between the two groups of patients.

In supplementary figure 13, several drugs are designated as “MEK inhibitor”, “BCL2 inhibitor”, etc; in figure 6, do the names of the therapeutic classes refer to the same drugs in the upper and the lower panel?

Overall, the conclusions of the authors related to the results of fig 6 are not supported by the results, notably their conclusions regarding the RAS mutational status and the benefit from MEK inhibitors

(line 608). The low number of patients included in this part of the work as well as the high heterogeneity of patients in terms of mutational status precludes any solid conclusion based on these results.

Minor: In supplementary figure 13, the scale should be identical for all patients.

Supplementary material

1/ A 2D plot of phosphoprotein expression by group (in particular p-ERK1/2 in the high- vs the low-24h-p-ERK1/2 group) is lacking in the supplementary data. This would enable to appreciate the quality of the staining, and would enable the readers to be more confident with the data currently presented as arcsinh transformed 90th percentile of p-ERK1/2.

2/ In supplementary table 1, the 24h-pERK1/2 status would be relevant information to include, which would enable to appreciate the repartition of the confounding factors (age, ELN, WBC count, and allogeneic stem cell transplantation and MRD) according to pERK1/2 expression.

Conclusion

The authors claim that "early single cell signaling response to chemotherapy provided precise prognostic information independent of stratification by genetics". Taking into account the limitations of this work ie the small sample size (N=32), the absence of effect on survival of validated prediction tools in this cohort (ELN), as well as the absence of comparison with the prediction based on the MRD, the authors may moderate this conclusion.

Minor comments and typos

Line 115: and overall poor overall survival

Line 618: the 18 patients that was analyzed

Reviewer #4:

Remarks to the Author:

The manuscript by Tislevoll et al. demonstrates an elegant new approach to predicting patient outcome based on molecular changes detected early in the treatment phase, rather than relying on genetic parameters only. The potential impact of such discoveries is vast, as it allows much more efficient treatment strategies to be determined and/or altered along the way if needed, thereby saving actual lives. Therefore I commend the authors for undertaking this work, which I expect will inspire many follow-up studies pursuing similar early treatment response markers across a range of diseases.

The manuscript is well written, the analyses conducted were thorough and nicely presented, and the combination of a range of technologies (and modalities) significantly strengthened the work as well in my opinion. The authors demonstrate appropriate use of the various technologies, and I especially liked the barcoding and reference sample spike in approach for their CyTOF analyses. Seeing a combination of single cell proteomics (on more than 35 million individual cells!), bulk proteomics and RNAseq, and targeted sequencing, in a clinical setting and all integrated through machine learning, is not a mean feat, and the results presented speak for themselves.

Thus, in my opinion, the work should be accepted for publication, and I have only a few minor comments laid out below:

1. On pg. 7, the authors declare the cells in MC1 and MC2 to be CD34+ blasts due to their high expression of CD34 and CD117. As these are also classical stem cell markers, could the authors clarify

why they are deemed differentiated blasts rather than a more primitive cell type?

2. Fig2b - is this average expression across the single cells measured within cluster MC9? Was any sub-segregation possible to determine the critical cell type in which response needs to be measured? In other words, given the single cell nature of their analyses, was there anything more specifically known about which cell types had high or low ERK? Or is it truly a homogeneous cell effect?

3. Regarding the RNAseq vs MS (Super-SILAC) results -> were none of the RNA observations confirmed in proteomics? I.e. were those candidates not detected in MS or did they show different results? A bit more discussion on the value of bulk MS vs the single cell CyTOF would also help strengthen the reasoning for including both in the study, and help follow-up work to decide for one or the other vs both.

4. I would like to request more detail on the in vitro drug screening experiments. E.g. what media was used, any growth factors or stromal cells, etc). Can they also comment on overall viability of the patient samples once put in vitro? As primary AML is notoriously difficult to culture in vitro while maintaining their hierarchical nature, it would be nice have this explored in a bit more detail. Their results were very encouraging, and could suggest potential alternative treatment strategies for those patients not responding to standard-of-care.

REVIEWER COMMENTS

Reviewer #1

expertise in AML genomics (Remarks to the Author):

In this manuscript, the authors used mass cytometry to measure the abundance of 36 extracellular and intracellular proteins in the peripheral blood of 32 AML patients at 0, 4, and 24 hours after treatment with 7+3 chemotherapy. Using LASSO Cox regression, p-ERK1/2 and p-p38 abundances in one myeloid cell population (called here “MC9”) at one time point (24 hr) were found to be significantly correlated with survival. This paper is well written and clear. The overall result is statistically significant and interesting, and suggests that this proteomic approach may become very important to the field. That result does not appear to be extremely robust given the arbitrary nature of the clustering analysis, and the lack of correlation with chemotherapy response, ELN risk status, transplantation status, or mutational status. As the authors note, this may simply be due to the size of the cohort, which is small relative to the heterogeneity and complexity of this disease. The authors also demonstrated that p-ERK1/2 and p-p38 abundance in flow-sorted blasts is correlated with survival, which strengthens the conclusions.

Specific comments:

1 MC9 is surprisingly heterogeneous with respect to CD34 expression. Because clustering is arbitrary, and clusters do not represent an absolute “ground truth,” this heterogeneity warrants further investigation for instance, given the heterogeneity of this important marker, can the authors justify calling this one cluster?

1.1 This is an important point and should always be a concern when analyzing CyTOF data. The different clustering algorithms available will assess the combination of surface markers and their expression levels to cluster similar cells together. The clustering will always depend on the combination of antibodies in the panel, and as the antibody number is limited, clusters will always be more or less heterogeneous also in respect to expression of surface proteins not included in the

panel. Furthermore, most algorithms are affected by manual input of different analysis parameters. For example, FlowSOM uses a manually set number of output clusters. Thus, a prerequisite for applying this algorithm is to have extensive biological knowledge of the sample analyzed, so one can assess approximately how many clusters one would expect to get out with the specific panel used. However, the heterogeneity of AML makes it difficult to decide on a specific number of clusters. One could choose a high number of clusters, which we have experimented with. This resulted in a high number of unique clusters represented by few patients, which is a challenge when performing cross-sectional analyses of an entire cohort. However, what we realized when experimenting with cluster number was that the (presumably) healthy cell subsets in the AML samples (i.e. lymphoid cells) always ended up in quite homogeneous clusters, whereas the myeloid cells formed heterogeneous clusters (Supplementary figure 2) This is likely due to aberrant expression levels of surface proteins that are more or less functional in the myeloid lineage, as opposed to healthy cells where the combination and expression levels of the surface proteins is tightly regulated and reflects cell function. We took advantage of this feature by selecting a cluster number that was sufficient to identify the expected healthy cell subsets (guided by healthy PB and BM samples) while clustering the remainder of the (presumably immature) cells in as few clusters as possible (in essence “under-clustering” the patient samples). Using this approach, we ended up with two blast clusters, one myeloid cluster and one stem cell cluster, where all patients were represented, which allowed a cross-cohort analysis of the signaling parameters.

The points above aptly illustrate, as rightly pointed out by the reviewer, that no cluster will ever represent an “absolute ground truth”. Clustering of single cell data is always a compromise. But based on this reasoning, we believe that we can justify calling MC9 one cluster. However, we certainly agree that MC9 is a heterogeneous cluster. Therefore, we have taken the concerns of the reviewer into consideration, and included a further characterization of the immunophenotype of MC9 in the revised manuscript (as more thoroughly described in the answer to question 2 below). We have also included a justification of the (slightly unconventional) analytical approach in the results section of the revised manuscript. (line 342)

2 In general, MC9 should be better characterized, for example, by highlighting each patient on the tSNE plot, and by showing the expression of each analyte, particularly CD34 and p-ERK1/2, on the tSNE plot, so that we can better understand heterogeneity within that cluster.

2.1 We thank the reviewer for this comment and have done a further characterization of MC9 in the revised manuscript, section “Immunophenotypical characterization of metacluster(MC9)”, new supplementary figures 7 and 8. We exported MC9 as identified by FlowSOM for all patients and did a new FlowSOM with only MC9. (Supplementary Fig. 9) We chose the same analytical approach as the original pipeline, with 10 metaclusters. These were named sub-clusters (Sub-MC1-10). As expected, the majority of cells clustered together (sub-C3 61%). We also performed a new LASSO Cox regression analysis, as described originally, using the 10 sub-Cs identified within MC9. This analysis confirmed the prognostic significance of pERK1/2 at 24h, but within a small sub-cluster; sub-MC7 (3.24% of total). This cluster was present, but small, in all patient samples. p-p38 was found to be of prognostic significance at 24h within a separate small sub-cluster; sub-MC2 (4.78% of total).

Supplementary Figure 8

2.2 We also manually gated the pERK1/2 positive and negative cells in MC9 separately in each patient, to investigate the immunophenotype of these cells. (Supplementary Fig.8) This analysis has been added to the result section of the manuscript under the paragraph "Immunophenotypic characterization of metacluster (MC) 9". Both positive and negative pERK1/2 cells had a heterogenous immunophenotype, but we found the expression of AXL($p=0.0002$), CD90 ($p=4.7E-05$), CD56 ($p=0.0006$) and CD34 ($p=0.002$) to be significantly higher in pERK1/2 positive cells.

2.3 We have made new tSNE plots that highlight the distribution of each of the 32 AML patients (identified by color) on the tSNE plot of all patients (grey) in MC9 Supplementary Fig. 22). These tSNE plots show only MC9. Supplementary figure 22a show all patients by color, b shows the 10 different sub-metaclusters as described in supplementary Fig 8b. And c shows the Individuals for each barcodepool, to make the distribution per patient more visible and to be able to assess if there are any batch effects. This figure has been added to the

supplementary information.

Supplementary Figure 22

2.4 Furthermore, we also made new tSNE plots showing the expression of pERK1/2 and CD34 in the different sub-clusters of MC9 (This figure is not included in the revised manuscript, but shown below (Response letter figure 2.4a)). These plots contain all 32 AML patients. Response letter figure 2.4b shows all surface markers in our antibody panel in MC9 at timepoint pre-treatment for all 32 AML patients. Notably, The FlowSOM analysis (described in 2.1) was performed on all cells in MC9 from all patients without down-sampling. The t-SNE plots (described in 2.1 and 2.4) are down-sampled equally per timepoint, showing 20.000 cells per plot.

Response letter figure 2.4

2.5 To estimate if the CD34 cells contributed to the prognostic effect of p-ERK1/2 in MC9, we exported MC9 for each patient and manually excluded/gated out the CD34 positive cells (response letter figure 2.5 a). Next, we assessed only the CD34 negative cells in MC9 and stratified the patients based on the 90th percentile 24h pERK1/2 value and divided the patients into two groups by the median 24h pERK1/2 value. There was still a significant difference in survival between the two groups (Response letter figure 2.5 b). (Log-rank (Mantel-Cox) test, p-value =0.0104)

Response letter figure 2.5

3 What proportion of malignant cells does MC9 represent in each patient?

3.1 The proportion of MC9 per patient is shown in Supplementary Figure 3 (red). In this figure, the proportion of this cluster is also shown for peripheral blood (yellow) and bone marrow (green) from healthy donors. MC9 is expanded in most of the AML patients compared to the 7 healthy donors. MC9 is more abundant in the three bone marrow samples than in peripheral blood, which indicates a more immature origin of these cells. We have assumed that this cluster is likely malignant due to the expansion in the AML patients and the immature myeloid phenotype. As described in 2.1 and shown in supplementary figure 8, we performed a new FlowSOM analysis on only MC9, identifying 10 sub-clusters (sub-C1-10). The immunophenotype of these sub-Cs for the 32 AML patients at diagnosis are shown in supplementary figure 8b. The largest sub-C, sub-C3

contained 61.07% of the cells in MC9, this cluster had a normal myeloid phenotype with C064, C038 and C033. Three clusters (sub-C 1, sub-C4 and sub-C2) had expression of C034 and were therefore classified as malignant cells (23.81%). The other sub-Cs had myeloid markers (C038, C033, C064) with aberrant lymphoid markers (C03, C04, C08, C020) and were therefore classified as aberrant clusters (15,12%). In the figure below, response letter figure 3.1, the malignant sub-clusters sub-C1, 2 and 4 was summarized and are shown as % of total MC9 cell population per patient.

Response letter figure 3.1

4 Does this proportion correlate with survival?

4.1 The size of each metacluster was included as a feature in the LASSO Cox regression analysis and was not predictive of survival. We also performed a new LASSO Cox regression analysis with only MC9 and the 10 sub-clusters. The size of neither of the sub-clusters was predictive of survival.

4.2 To investigate if the proportion of malignant cells correlated with survival, we stratified the patients based on % of malignant cells in MC9 and divided them into two groups by the median value (16 patients in each group). Response letter figure 4.2 (not included in the revised manuscript) shows the Kaplan Meier curve of the two groups. There was no significant difference between patients with a high proportion of malignant cells in MC9 and the ones with a low proportion (not significant, Log-rank (Mantel-Cox) test).

Response letter figure 4.2

5 Does change in the size of MC9 or time relate to survival?

5.1 No, the change (ratio 24h/0h) or the delta (24h-0h) of MC9 does not relate to survival. A simple linear regression showed that neither the delta nor the ratio of

MC9 was significantly correlated to survival. When we divided the patients into two groups (16 patients in each) and did a Kaplan Meire survival analysis, there was no significant difference. (Response letter figure 5.1. Not included in the revised manuscript).

Response letter figure 5.1

6 Does this depend on the extent of CD34 expression within MC9?

6.1 Only 10 of the 32 patients have CD34 expression in MC9, this is shown in supplementary figure 10b. Most of these patients (7/10) were in the high 24h-p-ERK1/2 group. The median level of CD34 was not significantly higher in high 24h-p-ERK1/2 group than in low group (grouped students t-test) and not when used as a categorical variable with 3 patients in low group and 7 patients in high group (Fishers exact t-test). The Kaplan-Meier survival analysis above (response letter figure 4.2) shows that when patients were stratified into two groups based on the proportion of CD34 cells in MC9, there was not a significant difference in survival between patients with high and low proportion of CD34+ cells in MC9. (Log-Rank Mantel cox test).

7 One limitation of the proteomic approach is that it is difficult to know exactly which cells are malignant, because no genetic/mutational information is available, although sorting and targeted sequencing for somatic mutations could be performed to address this further.

7.1 We agree that the lack of genetic information is a limitation for both the bulk proteomics and single cell CyTOF analyses. Sorting and targeted sequencing could certainly have been performed, although in our opinion this would not provide additional information that would be useful for assessing the results from proteomic analyses, as these are still from a bulk sample. It could, however, provide useful additional information for the CyTOF analyses. To our knowledge, no direct correlation between genotype and immunophenotype has been identified in AML, and such experiments could provide additional information to elucidate this connection. Furthermore, novel techniques are also available that can provide both genetic- and proteomic information from the same single cell analysis, such as CITE-seq (Stoeckius et al. Nature Methods 2017). Perhaps this could be a fitting

future approach to investigate this issue. However, this was outside the scope of the current work, and we did not have additional patient material available for such analyses.

- 7.2 *Although we don't have genetic information on the specific cell subsets identified by CyTOF, we still believe that identification of malignant cells is possible. Hematological diagnostics is heavily based on flow cytometry, where expansion of immature cells and/or cells with aberrant immunophenotypes is routinely identified as (likely) malignant. Thus, we think it is reasonable to assume that such cells identified by CyTOF are likely malignant as well. By including PB and BM samples from several healthy donors in our analyses, we ensure that cell populations with immature or abnormal phenotypes can be easily detected in the patient samples. Furthermore, the results from this study show that therapy response is not merely within the malignant subpopulation, and that presumably healthy subsets can also be involved in therapy response*
- 7.3 *Indirectly, cells in MC9 are more abundant in the three bone marrow samples than in peripheral blood, which indicates a more immature origin of these cells. We have assumed that this cluster is malignant due to the expansion in the AML patients and the immature myeloid phenotype. See also 3.1 for more details.*

8 Further discussion of the specific relevance of MC9 to survival would be helpful.

For example, why is protein expression in this cluster more relevant to survival than, say, the size or characteristics of the blast cluster, MC1?

8.1 *The relevance of altered protein expression in MC9 is identified through the Cox Lasso regression model. The model also included MC1 (and all other MCs) but did not identify significant alterations in these MCs following treatment that correlated to survival. As MC1 was a particularly interesting cluster due to the expression of CD34, this cluster has been further characterized in supplementary figure 10 and in the section "Signaling in CD34+ MC1 AML blast cluster" in the manuscript (line 385). Notably, pERK1/2 did not confer prognostic information in MC1. Why MC9 is relevant and not the clusters with typical blast immunophenotype, such as MC1, is a very interesting question. However, based on our data, we can merely speculate. As shown by Levine et al (Cell 2015) the immunophenotype is not necessarily correlated to the signaling and function of the malignant cells. Levine et al showed that the immunophenotype and signaling in healthy donors were tightly coupled. However, in the AML samples the stratification of primitive and mature signaling had no association with the CD34 expression. Surface markers might not be a reliable proxy for the cell function and the expression of surface markers might be more fluent than what we normally acknowledge. We have added some thoughts around this in the discussion section of the revised manuscript (line 670)*

9 Is change in the relative size of MC1 and MC9 (over time) correlated with survival?

9.1 *We have investigated the correlation between the size of MC9 at all timepoints as shown in response letter figure 9.1 below. The correlation to survival was not significant when the cluster size was used as a continuous variable as shown by the simple linear regression. In the original LASSO cox regression analysis, the metacluster size was also used as a continuous variable, and no metacluster size was predictive of survival. However, when patients were stratified by MC9 size into high and low group (split by median) with 16 patients in each group. (3 patients did not have samples at 4 hours, therefore we used 14 vs 15 patients for the 4h analysis). Patients with more cells in MC9 at 24h had a significantly poorer survival*

than the patients with less cells in MC9 at 24h. This was not significant at 0h or 4h timepoints. This underscores the negative prognostic value of MC9 at 24h after start of treatment.

Response letter figure 9.1

9.2 There is no correlation between the size of MC1 at any timepoint or the change in MC1 from 0h to 24h and patient survival, as shown in response letter figure 9.2 below. The correlation was analyzed by a simple linear regression, and was not significant for any of them. Patients were also stratified according to MC1 size at all timepoints and analyzed by Kaplan-Meier survival analysis, as described above. There was no statistical significance in survival between patients with high and low levels of MC1 at any timepoint.

Response letter figure 9.2

10 High and low p-ERK1/2 patients were defined relative to each other, depending on whether they were above or below the median in this cohort. Is there an absolute protein abundance that can be used to determine whether new patients are “high” or “low?”

10.1 The level of p-ERK1/2 in the healthy donors might be used to determine if the patients are high or low in p-ERK1/2. In Supplementary Fig. 5b, the level of pERK1/2 in patients in high 24h-pERK1/2 group is shown next to the 7 healthy donors in

MC9(bone marrow in green, peripheral blood in yellow). The level of p-ERK1/2 in MC9 In the high 24h-pERK1/2 group at 24 hours were significantly higher than the levels in the 7 healthy donors. The median value of pERK1/2 in the 7 healthy donors were 0.49 (range 0.42- 0.63). Thereby any value above this might indicate a poor prognosis. Kaplan-meier survival curve with a cut off <0.05 gives 10 patients in low group and 22 patients in high group: p-value of: 0.0004.(Kaplan Meier survival analysis, Log-Rank Mantel cox test)

Response letter figure 10.1

If not, how would new patients be classified with respect to p-ERK1/2 abundance?

10.2 Based on future analyses of healthy donors and new AML cohorts, it is likely that we can create a normal range for p-ERK1/2 and p-p38. However, we question if not the response based on therapy (p-ERK1/2 ratio) may be a more robust marker for response. We search for new cohorts in clinical trials that can be analyzed prospectively. The multiparameter analysis of CyTOF and spectral flow cytometry should allow standard samples and calibration beads could be added.

The result would be stronger if there were an additional validation cohort that could be independently classified with respect to p-ERK1/2 and p-p38 abundance.

10.3 We appreciate this suggestion and are planning future control cohort experiments, ideally in a controlled phase III trial. This is outside of the scope of this proof-of-principle study. The material in this analysis is based on full blood samples added a special fixation before red cell lysis and processing. Such material is not available in other cohort to our knowledge. Cryopreserved material will not allow a similar analysis of signaling at baseline and 24 h after start of chemotherapy. However, several lines of evidence support our observation. A) Control analyses of other cellular metaclusters do not reproduce the same prognostic information. B) Our results from similar collections in chronic myeloid leukemia (CML) with targeted kinase inhibitor therapy suggest that the concept is robust. (Gullaksen et al. Haematologica 2017)

11 Mutation correlations were very anecdotal. Obviously, this analysis would benefit from more patients and exome or WGS sequencing.

11.1 This is a very timely issue, and the prognostic classification of AML based on genomics is based on thousands of patients, e.g. ELN 2017 genetic risk classification (Dohner et al. Blood 2017, Herold et al. 2020). Our analyses indicate stratification with higher precision. This may have important implications for better precision medicine in AML prognostication. This needs to be addressed in larger trials, e.g. Loweberg et al. Blood Adv 2021. However, a challenge is that

specially prepared material is required. Your comment motivates us to test your suggestion in a larger prospective AML trial in the HOVON/SAKK network.

11.2 *Although we agree that the mutation correlations are anecdotal, the clinical data provided is contemporary diagnostics in most academic centers in Europe, focusing on use in ELN 2017 genetic risk classification including ongoing validation and sub stratification. We hypothesized that some mutations should have essential impact on signaling early during chemotherapy, more like indicated in previous works using ex vivo stimulation (Irish et al. Cell 2004). However, we found a tendency for patients with the same mutations to have a similar signaling pattern, and this might be useful in future studies. As stated in the manuscript, more robust analyses of the correlation between mutational status and signaling would require a much larger cohort.*

The confirmatory flow cytometry analysis on pp. 14-15 actually gave a more significant result than the original mass cytometry clustering analysis. This strengthens the overall claim of the paper, but raises some interesting questions. How do the immature cells defined by this blast gate correspond to the FlowSOM clusters shown earlier?

11.3 *For the validation by manual gating, our strategy was to exclude lymphoid cell subsets (CD45 high) and granulocytes (CD66b high). We agree that this gating strategy does not represent the same cells as in MC9, as also the CD34 positive cells are also CD45 low, CD66low. However, the main objective for performing this analysis was to investigate whether the significant functional signaling response could be identified in a bulk sample analogous to a crude blast gate used by conventional flow cytometry. There were several reasons for this, including to bridge our CyTOF analyses with conventional flow cytometry, which is routinely applied in the clinic. Furthermore, these results are also more comparable to our validation experiments by proteomics and RNAseq, as these are bulk analyses performed on lymphoprepped patient samples (where neutrophils are removed). The DSRT analyses are also performed using lymphoprepped samples.*

11.4 **Are these flow-gated blasts in MC9 or MC1?** *As stated above, the manual biaxial gate likely includes both MC9 and MC1. To be sure that the prognostic pERK1/2 signal did not come from the CD34 positive cells in the manually gated CD45, CD66low population, we also did a manual gating of the bi-axial gated cells where we gated away the CD34 positive cells to see if the prognostic information of pERK1/2 was still there. The gating strategy is shown in Response letter figure 12.2a below, b shows the 90th percentile arcsinh transformed pERK1/2 values at all timepoints for these cells. The prognostic value of pERK1/2 in this cell population was still significant, as shown by the Kaplan-meier curve (Response letter figure 12.2 c, Log-rank (Mantel-Cox) test: p-value 0.0048.)*

Response letter figure 12.2

12 The authors could reanalyze the cytometry data in a manner analogous to the flow sorting – this might strengthen the result, or would at least add clarity by connecting these two analyses to each other.

12.1 *We apologize for any unclarities in the original text, but the manual biaxial gating of immature cells (CD45^{low}/CD66^{b-}) is indeed a re-analysis of the original CyTOF data. We performed these analyses on the original unclustered data, as an alternative approach to unsupervised clustering using FlowSOM. The gating strategy was aimed at capturing all malignant cells in the samples, to investigate whether the significant signaling response could be identified in an analysis of something comparable to a bulk (lymfoprepped) sample, as described above. We have reviewed the text of the original manuscript, and made edits to the results section (line 400) to make this more clear.*

13 For the bulk RNA-seq analysis, a more unbiased analysis could be useful.

13.1 **What other genes were differentially expressed between the p-ERK1/2 low and high patients?** *To address this question, we did a student's t-test with FDR cut-off at <0.05 between the low vs high 24h-pERK1/2 groups (all timepoints). This analysis was performed on all genes in our RNAseq bulk dataset (n=50.668 genes). When these results were clustered by hierarchical clustering (Euclidean distance) the high and low 24h-pERK1/2 groups formed two separate clusters. The three patients with FLT3-ITD mutations formed a separate cluster within the high 24h-pERK1/2 group. The result was 76 significant genes. (Supplementary figure 11) 40 of these genes were high in high 24h-pERK1/2 group, there among HOXA9, HOXA10, HOXA10-AS. HOXA9 have been reported by others to be a poor prognostic factor in normal karyotype AML (Collins et al. Oncogene 2016) The myeloid oncogene TRIB1 is a pseudokinase that interacts with MEK1 to enhance its phosphorylation of ERK1/2. Overexpression of Trib1 enhances HOXA9 induced*

leukemogenesis (Takashi et al, Blood 2010). This figure has been added to the supplementary information as supplementary figure 11.

Supplementary Figure 11

13.2 Moreover, if the samples are hierarchically clustered based on the bulk RNA-seq data alone, do they naturally cluster into the same groups (p-ERK1/2 low and high)? To assess this question, we did an unsupervised hierarchical clustering (Euclidean distance) of all genes present in more than 3 samples ($n=50.490$) at all timepoints in all patients ($n=14$ patients). The heatmap is shown in the figure below (response letter figure 14.2a). There is no obvious clustering of the patients into high and low pERK1/2 group. All patients except for P16 made their own sub cluster with the three samples from each patient in the same cluster. This confirms the heterogeneity of the patients. Response letter figure 14.2b shows the unsupervised clustering of the pERK1/2 and pp38 inducible genes. Here the three patients with *Inv(16)* clustered together.

Response letter figure 14.2

14 The bulk RNA-seq t-test comparison essentially assumes that MC9 is roughly the same size in all patients. A bulk analysis that takes into account the size of MC9 in each patient might be more sensitive at detecting genes of interest.

14.1 *The bulk analysis is a weighted sum of all the metaclusters. So even if we correct for the size of MC9, we would still be left with different mixes of all other clusters. And we cannot correct for all of them. Notably, the RNAseq samples are lymphoprepped, the granulocytes are not included. The bulk RNA-seq t-test was not correlating with the size of MC9. We interpret this as gene expression in dominating genes, not defined gene expression.*

14.2 *However, in an attempt to answer this question, we have performed an analysis where we weighted for the **genes** highly expressed in patients with a high proportion of MC9, rather than to weight the analysis based on MC9 size. We did a student's t-test between the two patients with most cells in MC9, namely P14 and P15 (over 80% of the cells in MC9) and the two patients with the lowest number of cells in MC9; P9 and P20 (with less than 20% of the cells in MC9). All of these patients were in high 24h-pERK1/2 group. We selected the genes that were significantly higher in P14 and P15 (n=3637, p-value <0.05). Next, we did a grouped student's t-test between the patients in high 24h pERK1/2 group vs the ones in low 24h-pERK1/2 group on these selected genes (p-value cut of <0.05). P14, P15, P20 and P9 were excluded from this analysis. The significant genes are shown in response letter figure 15.2d. Patients in high and low group clustered into separate clusters, with hierarchical clustering (Euclidean distance). We found several genes of interest among the genes that were significantly higher in high 24h-pERK1/2 group. There among MAPK11, the beta isoform of P38, and three genes known to activate c-Jun (MAP3K13, MAP4K3 and MAP4K5). RUNX1 and MEIS1 was also significantly higher in high 24h-pERK1/2 group. Response letter figure 15.2 shows the results from this analysis. 15.2a shows the size of MC9 in each patients (patients with RNAseq data have a black border). Figure 15.2b*

shows the pERK1/2 level of the four patients used for the first t-test to enrich for genes more abundant in P14 and P15. 15.2c shows the unsupervised clustering of these genes. 15.2d shows a heatmap of the genes that were significantly different between the low and high 24h-pERK1/2 group.

Response letter figure 15.2

15 The drug sensitivity matrix should also be hierarchically clustered by patient and compared to the proteomic data to discover other potential correlates of drug sensitivity in an unbiased fashion.

15.1 The analysis has been performed, and a new figure has been added to the revised manuscript (Supplementary Fig. 15). The selective drug sensitivity score (sDSS) for all patients were clustered by hierarchical clustering (Euclidean distance), we applied a sDSS cut-off at +/-5, drugs that did not have values above or below 5 were not included (Supplementary figure 15a).

15.2 We also performed a grouped students t-test between the patients in high versus low 0h and 24h pERK1/2 group on the sDSS (cut-off +/-5). As the sDSS data were only performed at time of diagnosis, we chose to compare it with the pERK1/2 value at time of diagnosis. Two drugs were significantly different between the two groups, namely the HSP90 inhibitor Tanespimycin and the

hypomethylating agent Azacitidine/Vidaza). Both had high sDSS in high 0h pERK1/2 group. The most significant drug that had a high sDSS in patients with high pERK1/2 at 0h, was Azacitidine (Vidaza)($p=0.0128$). When we compared the low vs high 24h-pERK1/2 group, the most significant drug with high sDSS in high pERK1/2 group was the Akt inhibitor MK-2206 2HCl ($p=0.026$). Supplementary Fig.15b show the results of the students t-test for the 0h pERK1/2 groups, patients are sorted by the pERK1/2 value in MC9 at 0h. Supplementary Fig.15c show the correlation between sDSS for Azacitidine vs pERK1/2 in MC9 at 0h. (Simple linear regression analysis, p -value=0.0159). This figure have been added to the supplementary information and described in the Paragraph “Drug sensitivity data show sensitivity for HSP90,mTOR, BCL-2 and MEK inhibitors”.

Supplementary Figure 15

16 The authors repeatedly mention that their proteomic data indicates the presence of “clonal heterogeneity,” but this is traditionally defined by genetics, not proteomics. For instance, clustering of somatic mutations by variant allele frequency would reveal clonal heterogeneity.

16.1 It is true that the link between genotype and immunophenotype is not well explored, and it is currently not known whether immunophenotypic heterogeneity in AML is related to clonal heterogeneity, although it is known that the disease is characterized by both. Thus, we agree that this term has been improperly used in the manuscript. This statement has now been removed throughout the revised manuscript. We have further deleted a paragraph in the discussion where clonal heterogeneity and the advantages of single cell analyses in this context was discussed.

Reviewer #2,

expertise in bioinformatics, mass cytometry analysis and AML (Remarks to the Author):

The authors used 36-dimensional mass cytometry data to identify genes sensitive to the chemotherapy response. These were further validated using RNAseq and mass spec. The reason of this study is well explained and further validated using DSRT.

18. The tSNE plot in Figure 1 can be colored with ID to see if clustering is not influenced by batch effect.

18.1 The issue of batch effect in mass cytometry data is an important question, and we are glad to elaborate on this in our data. Notably, to be able to correct for batch effects, we included an identical reference sample that was added to each of the 7 barcodepools (barcode 1-7) and used for normalization (quantile normalization). Thereby we were able to correct for the batch effects, and compare the different patients across barcodepools. A new figure has been added to supplementary Figure 16, (Supplementary figure 18e) to show the reference sample prior to and after normalization, the t-SNE plots illustrate how there was a batch effect prior to normalization and show how the batch effect is removed after normalization. CD4 and CD66b staining is shown in the normalized samples.

Supplementary figure 18e

19. Many clusters including MC1,2,3 are scattered. Isn't it because of batch effect? coloring based on patient ID will answer.

19.1 As suggested by the reviewer, we have made new t-SNE plots showing the different barcodepools (batches) and colored by patient ID. These figures have been added to the supplementary information as Supplementary figure 20. The data is not affected by batch effect in our opinion. However, due to the heterogeneity of the disease, there are some clusters that consist of few patients, like MC2, which is mainly consisting of P9 (batch 1) and P22 (batch 4). We have also added some text in the methods section to elaborate on potential batch effects (line 782) Supplementary figure 20a shows the 10 FlowSOM metaclusters. Figure b shows the 7 different barcodepools/batches. Figure c shows the 32 individuals and the 7 healthy donors.

Supplementary Figure 20

20. Why they are scattered? Is it due to the assignment to the existing clusters? How then they can be assigned to the same cluster? For instance, MC1 is scattered into to clusters? How these are defined as MC1?

20.1 FlowSOM consists of a clustering step that assigns cells to many clusters and a meta-clustering step that combines some of these clusters. The step of combining clusters can in some cases also result in meta-clusters that appear somewhat scattered on a t-SNE map. The visually scattered clusters indeed reflect phenotypic heterogeneity within the clusters identified by FlowSOM (e.g. variable surface marker expression levels). For instance, there are five known subsets of t-helper cells, and although specific markers to identify these subsets are not included in the antibody panel, the t-helper population will still have a slightly variable expression levels of lymphoid markers, which reflects different biological functions.

- Importantly, tSNE is a visualization tool, not a clustering tool. tSNE plots are used to visualize high-dimensional data in two dimensions. Its purpose is not to show clusters in data (<https://arxiv.org/pdf/2110.02573.pdf>). It would be surprising if there would be no correspondence at all between tSNE and clusters when they are both based on the same combination of surface markers, but we cannot expect complete overlap. - Clustering or gating on the tSNE-transformed data would lead to the least scattering on the tSNE, but it has been shown that this can result in less predictive clusters than clustering the high-dimensional data using unsupervised algorithms. We focused on the predictive power of our clustering rather than the visual appearance.

- FlowSOM was performed on all cells from all patients, without down sampling.
The t-SNE plots show 20.000 cells per plot.

21. MC9 has been extensively used for the analysis. Why MC9 (the myeloid cluster) could become most informative cluster among other MCs? Any speculation on it?

21.1 MC9 was chosen from the LASSO regression analysis. This is the metacluster where we find the prognostic value of pERK1/2 and p-P38 at 24h. As shown in supplementary figure 8, we performed a new FlowSOM analysis on only MC9, identifying 10 sub-clusters (sub-C1-10). The immunophenotype of these sub-Cs for the 32 AML patients at diagnosis are shown in supplementary figure 8b. The largest sub-C, sub-C3 contained 61.07% of the cells in MC9, this cluster had a normal myeloid phenotype with C064, C038 and C033. Three clusters (sub-C 1, sub-C4 and sub-C2) had expression of C034 and were therefore classified as malignant cells (23.81%). The other sub-Cs had myeloid markers (C038, C033, C064) with aberrant lymphoid markers (C03, C04, C08, C020) and were therefore classified as aberrant clusters (15,12%). When we did a new FlowSOM analysis on only MC9 we discovered that the prognostic pERK1/2 signal did not seem to originate from these C034 positive cells, but from a smaller myeloid but aberrant cluster (sub-cluster 7) which contained 3.24% of the cells from the 32 AML patients (Supplementary figure 8d). This is discussed in the new section in our manuscript "Immunophenotypic characterization of metacluster (MC) 9."

22. Besides 24h-p-ERK1/2, p-p38 at 24, p-Rb at 24h were reported. The combination of several markers could lead a better performance. This has to be tested and discussed. In the same way, can the information obtained from RNAseq and Mass spec be used to in designing combination of markers?

22.1 Note that the Lasso Cox regression model does find a combination of markers, namely the combination of p-ERK1/2, p-p38, and p-Rb at 24h. We have chosen to report on these individually, because this makes it possible to show Kaplan-Meier curves split by median of each individual marker. But the overall model uses a linear combination of the makers.

It is true that the Lasso regression does not include any interaction terms between the different markers. While this is in principle possible, it increases the number of regression coefficients from around 600 to around 180000. For this to be reasonable, we would require a substantially larger sample size.

22.2 The RNAseq and Mass spec are bulk cell analyses and therefore they are difficult to use to find combinations of markers that can be applied in single cell analysis.

Reviewer #3

expertise in mass cytometry and AML (Remarks to the Author):

In the present manuscript the authors assessed the correlation between single cell signaling and clinical response to induction chemotherapy in 32 patients with AML. Samples were collected at diagnosis and during the course of induction chemotherapy, and analyzed by mass cytometry, RNAseq and mass spectrometry proteomics in order to assess 1/ the chemotherapy-induced phospho-signaling modulations in the myeloid compartment and 2/ the prognostic significance of phospho-protein expression levels in myeloid cells 24h after treatment initiation. Overall, the manuscript is well written and the mass cytometry data are presented in a clear way. However

some issues detailed below prevent publication of this manuscript in its present form. Graphical abstract :

23. The legend is constructed as a standard legend; this legend should rather focus on the summary of the key points and the main conclusions of the article.

23.1 the graphical abstract has been reconstructed into figure 1.

24. The risk stratification by bone marrow tumor load is mentioned in the discussion section, but has not been displayed in the results section or in the supplementary data. The overall survival according to the ELN classification only appears in the graphical abstract as well, and should be integrated to the body of the article.

24.1 We thank the reviewer for this observation, the ELN risk stratification has now been described in the manuscript and is now displayed in the results section as Figure 1.

Results section

25. Identification of an independent prognostic factor in AML should be confirmed using a Cox regression including the validated prognostic factors in order to exclude potential confounding factors. In the present work, this Cox regression should include 24h-pERK1/2, age, ELN, WBC count at diagnosis, as well as allogeneic stem cell transplantation, ideally as a time-dependent covariate. Of note, the poor survival in the ELN favorable group, especially taking into account the fact that most patients are young patients, underlies that there might be a bias in the present cohort and probably precludes proper Cox regression for a formal identification of a potential prognostic factor.

25.1 We have performed a new cox regression analysis that has been added to the results section of the revised manuscript (line 258), which included the following covariates for all patients: The 24h pERK1/2 value in MC9 (continuous variable), age (continuous variable), ELN 2017 risk (categorical variable: Favorable, Intermediate, Adverse), WBC count at diagnosis (continuous variable) and transplantation status as a time-dependent covariate. There were 14 patients who received allogeneic transplantation in our study. All patients were followed for 5 years or until they died. No patients dropped out of the study.

The result of this analysis was:

parameter	hazard_ratio	p
1 pERK24h	2.27	0.000581
2 Age	0.0123	0.571
3 WBC at diagnosis	-0.00690	0.290
4 ELN	0.0227	0.941
5 transplant	-0.252	0.639

Thus, pERK1/2 value at 24h was the only predictive marker for patient outcome (5 year survival, HR 2.27, p-value 0.000581).

25.2 To make it easier to evaluate the composition of the patient cohort, we have added a new summary table of the cohort to the supplementary information. (supplementary table 2) Notably, these patients were consecutively sampled over 2 years at Bergen university hospital and Oslo university hospital. The only inclusion criteria was AML, undergoing standard 3+7 induction therapy.

26. The MRD status is a cornerstone for prognostication of AML. An analysis with respect to the MDR would therefore be essential before concluding about the potential usefulness of measuring pERK1/2 to refine the therapeutic decision algorithm in AML.

26.1 MRD data was available for 14 of the patients in this study. Seven patients in low 24h-pERK1/2 group and seven in high pERK1/2 group. 5/7 patients in low group had negative MRD after cycle 2, the remaining two had no detectable leukemia associated immunophenotype (LAIP) at diagnosis (P10) and the other one had negative MRD but positive NPM1 MRD (P13). Among the seven patients in high group, four had negative MRD and three patients had positive MRD. The three patients with positive MRD were among the six patients with highest pERK1/2 values in MC9 at 24h after start of therapy. We found no significant differences in MRD status between patients in the high and low pERK1/2 group. The MRD data has now been added to the patients characteristics table, figure 3b and is described in the paragraph "Clinical parameters related to the p-ERK1/2 level in MC9" in the revised manuscript.

27. The authors claim in the abstract that they questioned whether the signaling response to therapy was more informative than analysis at time of diagnosis.

In fig 2d, the basal level of pERK1/2 seems low in the low-24h-pERK1/2 group and high in the high-24h-pERK1/2 group.

Did the authors assess the impact of baseline expression in the metacluster 9 on clinical outcome for a direct comparison?

This information is important in order to appreciate the improvement of the quality of prediction using samples drawn during induction chemotherapy rather than baseline samples.

27.1 These analyses were done, and we found no significant correlations with differences in baseline expression and survival. This has been added to the results section of the revised manuscript.

The 0h timepoint (in all metaclusters) is included in the LASSO cox regression and was not significant.

Additionally, when patients were stratified by 0h pERK value and divide into two groups (with 16 patients in each group) based on median pERK1/2 at 0h value, the survival between the two groups is not significant.

Response letter figure 27.1

28. Why did the authors choose to include patients with MDS and B-ALL in this manuscript? At least the authors should retain only the 32 patients included in the study in the patient characteristics table.

28.1 We agree with the reviewer, these patients were not used in any of the presented analysis and have now been removed from the manuscript. We have chosen to keep the two patients P33 and P34 who received dose reduced 3+7 treatment. In a separate analysis (FlowSOM clustering and LASSO cox regression analysis) together with the other 32 patients, we still found a significant association with survival and pERK1/2 and pp38 at 24 hours. This is described in the methods section and strengthens the stability of the LASSO cox regression analysis.

29. In supplementary fig 3, the cluster of NK cells is located at 2 different places on the tSNE plot. Some cells that were annotated as NK cells clusterize in the region of myeloid cells, and might be leukemic blasts that express CD56. This would also explain the low CD45 mean expression of this cluster compared with CD45 expression in the clusters of CD4 and CD8 T cells. The authors should confirm by manual gating the exact nature of these cells.

29.1 To define the exact nature of these cells, we exported the NK cell metacluster (MC8) and did a new t-SNE analysis on only the NK cell cluster. These t-SNE plots show MC8 for all patients at the three different timepoints (all MC8 cells from each patient were concatenated at each timepoint) and for the 7 healthy donors. The t-SNE plots show the expression of all surface markers. From these plots there is an obvious difference between the patient samples and the healthy donors. One island on the tSNE plot is not present in the healthy donors, this island is positive for CD33, CD56, negative for CD7 and have a lower CD45 expression. We agree with the reviewer that these cells might be leukemic blasts that express CD56.

30. The authors should provide expression of the different markers projected on the tSNE plots in supplementary 3 in order to enable appreciation of the homogeneity of the different clusters. Minor: in supplementary fig 3, the lack of heatmap annotation in the lower panel makes the figure difficult to read.

30.1 A supplementary figure showing the expression of each surface marker on the tSNE plots have been added to the revised manuscript (Supplementary figure 19).

We have added the annotation on the lower panel of heatmaps in supplementary figure 3 (this figure is supplementary figure 2 in the revised supplementary information).

Supplementary figure 19

31. In Fig 3, the authors detail the manual gating strategy of leukemic blasts in order to confirm the results obtained with machine learning algorithms and assess the transferability of the results in a routine setting. However, the metacluster used to generate the initial survival curves according to pERK1/2 expression was MC9, which was considered as a myeloid cluster, by contrast with the MC1 and MC2 that were considered as the AML blast clusters (line 198). Could the authors clarify this discrepancy in the selection of the cell population they chose for manual confirmation of the results?

31.1 This is an important point, and we thank the reviewer for pointing this out. As further elaborated in the revised version of the manuscript, MC9 is a very heterogeneous cluster, and identification of similar/comparable cells in single AML samples by flow cytometry would not be feasible. Thus, the reason for this specific gating strategy was rather to investigate whether a gating for all immature/leukemic blasts cells would provide comparable information. In addition to assessing the transferability to routine flow diagnostics, another reason for investigating this population specifically was to increase the comparability between the CyTOF data and the validation datasets, consisting

of bulk proteomics and RNAseq data, since the samples used for proteomics and RNAseq were density centrifuged mononuclear cells and thus depleted of granulocytes.

32. In fig 6, results are presented for 16 and 9 drugs in the low 24h – p-ERK1/2 group and the high 24h – p-ERK1/2 group, respectively. The lack of consistence between panel b and c makes it difficult to compare the graphs. In addition, the drugs that are displayed in figure 6 should be listed in the materials and methods section (“MEK inhibitor”, “DNA inhibitor” or “anti-oxidant” being too evasive). The barplots should be annotated so as the drug class appears as “Protein X inhibitor” rather than “Protein X”.

32.2 We have revised figure 6 (called figure 7 in the revised manuscript) according to the reviewers’ suggestions. We chose to visualize the 12 most efficient drugs in the entire cohort for both high and low 24h-pERK1/2 group (as shown in supplementary figure 16a). The same drug targets are visualized for both groups to make the two groups more comparable. We have specified that the different targets are inhibitors in the figure legend. We have added the specific drug names of the different drug target groups per patient in a supplementary excel table (Supplementary table 6) (sheet 2 “top 10 drugs per patient”) together with the sDSS data of all patients (Sheet 1 “sDSS”).

33. The aim of this figure is to correlate the intracellular signaling profiles detected by mass cytometry with ex vivo drug sensitivity (line 573). The way the results are presented does not enable a direct comparison between the two groups of patients.

33.1 We have rearranged the data to improve readability (figure 7 in the revised manuscript)

33.2 We have performed a new analysis where we do hierarchical clustering of the sDSS data comparing it to the pERK1/2 groups (Supplementary figure 15a) The selective drug sensitivity score (sDSS) for all patients were clustered by hierarchical clustering (Euclidean distance), we applied a sDSS cut-off at +/-5, drugs that did not have values above or below 5 were not included (Supplementary figure 15a). We also performed a grouped students t-test between the patients in high versus low 0h and 24h pERK1/2 group on the sDSS (cut-off +/-5). As the sDSS data were only performed at time of diagnosis, we chose to compare it with the pERK1/2 value at time of diagnosis. Two drugs were significantly different between the two groups, namely the HSP90 inhibitor Tanespimycin and the hypomethylating agent Azacitidine/Vidaza). Both had high sDSS in high 0h pERK1/2 group. The most significant drug that had a high sDSS in patients with high pERK1/2 at 0h, was Azacitidine (Vidaza)(p=0.0128). When we compared the low vs high 24h-pERK1/2 group, the most significant drug with high sDSS in high pERK1/2 group was the Akt inhibitor MK-2206 2HCl (p=0.026). Supplementary Fig.15b show the results of the students t-test for the 0h pERK1/2 groups, patients are sorted by the pERK1/2 value in MC9 at 0h. Supplementary Fig.15c show the correlation between sDSS for Azacitidine vs pERK1/2 in MC9 at 0h. (Simple linear regression analysis, p-value=0.0159). This figure have been added to the supplementary information and described in the Paragraph “Drug sensitivity data show sensitivity for HSP90, mTOR, BCL-2 and MEK inhibitors”.

34. In supplementary figure 13, several drugs are designated as “MEK inhibitor”, “BCL2 inhibitor”,

etc; in figure 6, do the names of the therapeutic classes refer to the same drugs in the upper and the lower panel?

34.1 To improve interpretability of this data-intensive figure, we have annotated the figure using drug class (based on drug target) instead of using the specific drug name. This allows for an easier biological interpretation of the results. However, we appreciate that it could be interesting for some readers to compare the effect of specific drugs side-by-side, and we have included the drug names for the different drug targets for each patient in supplementary table 6, sheet 2.

35. Overall, the conclusions of the authors related to the results of fig 6 are not supported by the results, notably their conclusions regarding the RAS mutational status and the benefit from MEK inhibitors (line 608). The low number of patients included in this part of the work as well as the high heterogeneity of patients in terms of mutational status precludes any solid conclusion based on these results.

35.1 We apologize if this was poorly explained, the RAS mutation data comes from supplementary 14, where the other 2 patients with RAS mutations are shown, not from Figure 6. However, as these patients were not included in the main LASSO cox regression analysis we have chosen to remove this figure and the description of it in the revised manuscript. We have toned down our conclusion regarding the benefit from MEK inhibitors.

36. Minor: In supplementary figure 13, the scale should be identical for all patients.

36.1 The scale is now made identical for all patients. (Supplementary figure 16 in revised manuscript)

37. 2D plot of phosphoprotein expression by group (in particular p-ERK1/2 in the high- vs the low-24h-p-ERK1/2 group) is lacking in the supplementary data. This would enable to appreciate the quality of the staining, and would enable the readers to be more confident with the data currently presented as arcsinh transformed 90th percentile of p-ERK1/2.

37.1 A new figure has been added to the supplementary information of the revised manuscript (Supplementary figure 23). We exported MC9 for each patient and healthy donors. Then we did a new tSNE with all cells for each patient at the three timepoints and the bone marrow or peripheral blood from the healthy donor that were in the same barcode as that particular patient. These tSNE plots are shown in supplementary figure 23. Patients are stratified based on their 24h pERK1/2 value in MC9, from low to high. Patients in low 24h-pERK1/2 group are in Supplementary

figure 23 a, patients in high 24h pERK1/2 group are shown in b.

Supplementary figure 23

38. In supplementary table 1, the 24h-pERK1/2 status would be relevant information to include, which would enable to appreciate the repartition of the confounding factors (age, ELN, WBC count, and allogeneic stem cell transplantation and MRD) according to pERK1/2 expression.

38.1 *We thank the reviewer for this helpful suggestion. pERK1/2 status has now been included in the Patients characteristics table (supplementary table 1)*

38.2 *We have also done statistical analysis of the clinical data between the two groups. Age, ELN, WBC or allogeneic stem cell transplantation were not significantly different between the high and low 24h-pERK groups. This has been added and further described in the section “Clinical parameters related to the p-ERK1/2 level in MC9” in the revised manuscript.*

We also performed a new cox regression analysis where we included pERK1/2 in MC9, age, ELN, WBC at time of diagnosis and allogeneic transplantation as a time-dependent covariate. Only pERK1/2 was significant in predicting patient 5 year survival (HR 2.27, p-value 0.000581)(Line 258) in the revised manuscript.

Conclusion

39. The authors claim that “early single cell signaling response to chemotherapy provided precise prognostic information independent of stratification by genetics”. Taking into account the limitations of this work ie the small sample size (N=32), the absence of effect on survival of validated prediction tools in this cohort (ELN), as well as the absence of comparison with the prediction based on the MRD, the authors may moderate this conclusion.

39.1 *We agree with the reviewer, and the discussion and conclusion of the revised manuscript has been re-written to moderate/tone down this conclusion (Line 690)*

Minor comments and typos

Line 115: and overall poor overall survival

Thank you, this has been corrected

Line 618: the 18 patients that was analyzed

Thank you, this has been corrected

Reviewer #4

expertise in proteomics/super-SILAC (Remarks to the Author):

The manuscript by Tislevoll et al. demonstrates an elegant new approach to predicting patient outcome based on molecular changes detected early in the treatment phase, rather than relying on genetic parameters only. The potential impact of such discoveries is vast, as it allows much more efficient treatment strategies to be determined and/or altered along the way if needed, thereby saving actual lives. Therefore I commend the authors for undertaking this work, which I expect will inspire many follow-up studies pursuing similar early treatment response markers across a range of diseases.

The manuscript is well written, the analyses conducted were thorough and nicely presented, and the combination of a range of technologies (and modalities) significantly strengthened the work as

well in my opinion. The authors demonstrate appropriate use of the various technologies, and I especially liked the barcoding and reference sample spike in approach for their CyTOF analyses. Seeing a combination of single cell proteomics (on more than 35 million individual cells!), bulk proteomics and RNAseq, and targeted sequencing, in a clinical setting and all integrated through machine learning, is not a mean feat, and the results presented speak for themselves.

We are grateful for these very encouraging remarks.

Thus, in my opinion, the work should be accepted for publication, and I have only a few minor comments laid out below:

40. On pg. 7, the authors declare the cells in MC1 and MC2 to be CD34+ blasts due to their high expression of CD34 and CD117. As these are also classical stem cell markers, could the authors clarify why they are deemed differentiated blasts rather than a more primitive cell type?

40.1 MC1 and MC2 were much more abundant in AML samples compared to healthy donors as shown in supplementary figure 3 (revised manuscript), and was only detectable in healthy bone marrow at very low levels (<0.5%) and in healthy peripheral blood in even lower levels. MC1 and MC2 had also aberrant marker expression like CD7 and CD56, we therefore classified these cells as CD34+ AML blasts.

40.2 We have annotated the CD34 positive metaclusters MC1 and MC2 for CD34+ cells blasts and not leukemic stem cell as we think that these terms may be controversial. There is an ongoing discussion about the definition of leukemic stem cells based on the immunophenotype (Khaldoyanidi SK et al. Crit Rev Oncol Hematol. 2022; Vetrie D, Helgason GV, Copland M. Nat Rev Cancer. 2020), and we have therefore chosen a broader term by calling these cells CD34+ blasts.

41. Fig2b - is this average expression across the single cells measured within cluster MC9?

41.1 The heatmap shows the Arcsinh transformed 90th percentile value of pERK1/2 in MC9.

42. Was any sub-segregation possible to determine the critical cell type in which response needs to be measured? In other words, given the single cell nature of their analyses, was there anything more specifically known about which cell types had high or low ERK? Or is it truly a homogeneous cell effect?

42.1 We think this is a crucial question and have done additional analysis to address this. To identify the cells from where the pERK1/2 signal originated from we did a new FLOWSOM with 10 metaclusters of only the cells in MC9 for all patients. A new LASSO cox regression analysis of these 10 sub-clusters identified that ERK1/2 and pp38 originated from two separate smaller clusters within the original MC9 (Sub-cluster 2 and Sub-cluster 7). These new findings have been described in a new paragraph in the revised manuscript "Immunophenotypical characterization of metacluster (MC) 9" and new supplementary figures have been added to the supplementary information (supplementary figure 7 and 8). When we investigated the immunophenotype per patient in Sub-C7 there is still a heterogeneous expression of surfacemarkers among the 32 AML patients (Response letter figure 42.1). We assume that the malignant clusters will never obtain the same homogeneity as we see in normal healthy cells (healthy cell clusters in supplementary figure 2). This is due to the heterogeneity of AML and one of the defining factor of malignant cells; they have aberrant surface marker expression. The surface markers might not be the best proxy for cell function, and they are expressed in a continuum over the course of the hematopoietic hierarchy. This continuum is very difficult to capture with clustering approaches and this

forces the cells into artificial clusters/metaclusters. Therefore, we chose an approach which underclusters the data and focused on the signaling.

42.2 To address the exact immunophenotype of cells with high pERK1/2 in MC9 we did a manual gating strategy where we gated the positive and negative pERK1/2 cells in MC9 for each patient. The results are shown in the new supplementary figure 7. Both the pERK1/2 positive cells and the negative had a heterogenous immunophenotype. AXL, CD90 and CD56 were significantly higher in pERK1/2 positive cells (grouped students t-test) and when we did a paired students t-test CD34 was also significantly higher in pERK1/2 positive cells. This has been described in the revised version of the manuscript (line 310).

Supplementary figure 67

43. Regarding the RNAseq vs MS (Super-SILAC) results -> were none of the RNA observations confirmed in proteomics? I.e. were those candidates not detected in MS or did they show different results? A bit more discussion on the value of bulk MS vs the single cell CyTOF would also help strengthen the reasoning for including both in the study, and help follow-up work to decide for one or the other vs both.

43.1 We have investigated if any of the significant genes in our RNAseq analysis could be detected in the proteomics. For the genes shown in figure 5d, only HSP90AA1 could be detected, the other ones were not detectable in our MS data. Among the AP-1 family genes shown in supplementary figure 13b, neither FOLS1 or ATF3 could be detected in the proteomics, neither could FOS, FOSB, ATF3, JUN or MAFF. FOSL-2 was found in the proteomics data but only in 7/15 patients. JUNB was only present in 6/15 patients. JUND was the only member of the AP-1 family that were present in most patients in our

proteomics data. (Response letter figure 43.2)

Response letter figure 43.2

43.3 In our unsupervised analysis between the 24h low and high pERK groups with an FDR cut-off at 0.05 (new supplementary figure 11), we found 76 significant genes in our RNA seq data. We did the same with our proteomics data, low vs high 24h-pERK1/2 group with an FDR cut off at 0.05 and found 193 significant proteins. IGF2BP2 (Insulin Like growth Factor 2 MRNA binding protein 2) was the only gene/protein that was significant in both the proteomics and RNAseq data analysis. It was high in high group in both the RNAseq data and the proteomics data. IGF2BP2 is an m6A reader gene and have been shown to have a negative prognostic value in AML.

44. I would like to request more detail on the in vitro drug screening experiments. E.g. what media was used, any growth factors or stromal cells, etc). Can they also comment on overall viability of the patient samples once put in vitro? As primary AML is notoriously difficult to culture in vitro while maintaining their hierarchical nature, it would be nice have this explored in a bit more detail. Their results were very encouraging, and could suggest potential alternative treatment strategies for those patients not responding to standard-of-care.

44.1 We apologies that the methodology of the DSRT experiments were not described in sufficient detail. A more detailed description of the methodological approach is added to the methods section of the revised manuscript line(853). MCM media was used, supplemented with 1%PS, no growth factors or stromal cells were used. The viability of the patient samples in vitro has not been assessed.

Reviewers' Comments:

Reviewer #1:

Remarks to the Author:

The authors addressed my concerns in great detail. I am still troubled by the fact that it is not conclusively shown that the relevant cells in MC9 are malignant, but the authors' inference does seem reasonable.

Reviewer #2:

Remarks to the Author:

My questions are well answered.

Reviewer #3:

Remarks to the Author:

A very good quality review was carried out by the authors. All the issues raised have been addressed in this revised version and have led the authors to make changes that are definitely adapted to the weaknesses raised. The limitations of the work have been discussed; conclusions that are not supported by the data do not appear anymore. However, there is still a problem with question 25. In the results of the multivariate analysis, hazard ratios are negative for the WBC count as well as allogeneic stem cell transplantation. This part must be reviewed carefully by a statistician.

Reviewer #4:

Remarks to the Author:

Tislevoll et al have clearly taken into consideration the comments of the reviewer collective, and significantly enhanced the quality of their manuscript. The additional analyses conducted, clarifications provided and especially the in-depth analysis of MC9 and what cell-types might exist therein, have all contributed to raising the impact of this manuscript even further.

From my perspective, all my concerns have been addressed and I would recommend publication of the manuscript in its current form.

Reviewer #5:

Remarks to the Author:

Here are my concerns for the statistical analysis and, in particular, survival analysis.

The biggest concern is that the study would be severely underpowered with only 32 AML patients, among whom 20 deaths were observed. With this sample size, it would be difficult to obtain an assertive statement that "Chemotherapy-induced changes in intracellular signaling during the first 24 hours of 233 treatment predicts long-term survival" as the obtained associations were likely to be spurious.

Second, all the p-values were without multiple comparison corrections and, as a result, the type I error (or more precisely the family wise error) could not be controlled.

Third, the machine learning results were not described clearly. How did you set up your training and validation samples? Did you do cross-validation? Normally, machine learning helps you generate hypotheses to test in future studies. I am not sure how machine learning strengthens your results here.

Response letter to reviewers

Reviewer #1 (Remarks to the Author):

1. The authors addressed my concerns in great detail. I am still troubled by the fact that it is not conclusively shown that the relevant cells in MC9 are malignant, but the authors' inference does seem reasonable.

1.1 We thank the reviewer for the positive feedback.

Reviewer #2 (Remarks to the Author):

2. My questions are well answered.

2.1 We are glad the reviewer found our revision to be satisfying.

Reviewer #3 (Remarks to the Author):

3. A very good quality review was carried out by the authors. All the issues raised have been addressed in this revised version and have led the authors to make changes that are definitely adapted to the weaknesses raised. The limitations of the work have been discussed; conclusions that are not supported by the data do not appear anymore. However, there is still a problem with question 25. In the results of the multivariate analysis, hazard ratios are negative for the WBC count as well as allogeneic stem cell transplantation. This part must be reviewed carefully by a statistician.

3.1 We thank the reviewer for this comment and sincerely apologize for our mistake regarding the calculations of the hazard ratio. The hazard ratios we have written in our manuscript were log hazard ratios. This explains the negative values. We have now added log to all hazard ratios in the revised manuscript.

Reviewer #4 (Remarks to the Author):

4. Tislevoll et al have clearly taken into consideration the comments of the reviewer collective, and significantly enhanced the quality of their manuscript. The additional analyses conducted, clarifications provided and especially the in-depth analysis of MC9 and what cell-types might exist therein, have all contributed to raising the impact of this manuscript even further.

From my perspective, all my concerns have been addressed and I would recommend publication of the manuscript in its current form.

4.1 We thank the reviewer for these encouraging words.

Reviewer #5 (Remarks to the Author): Expert in biostatistics

Here are my concerns for the statistical analysis and, in particular, survival analysis.

5. The biggest concern is that the study would be severely underpowered with only 32 AML

patients, among whom 20 deaths were observed. With this sample size, it would be difficult to obtain an assertive statement that "Chemotherapy-induced changes in intracellular signaling during the first 24 hours of 233 treatment predicts long-term survival" as the obtained associations were likely to be spurious.

5.1 We agree with the reviewer, the cohort size is unfortunately small. This is mainly because the patient material we present in this study is unique. Samples are collected shortly after start of treatment and processed using fix/lysis right after sample collection to preserve the phosphorylation status of intracellular signaling proteins. This requires immediate on-site sample processing by qualified personnel in appropriately equipped labs, which limits sample collection significantly. Additionally, AML is a relatively rare diagnosis, with approximately 150 new cases diagnosed per year in Norway (population: 5,4 million). Thus, to our knowledge, no other comparable datasets of AML patient material exist to date.

The small sample size was a major concern when deciding on an analytical approach to this dataset. Indeed, the selection of LASSO regression over many other machine learning algorithms (e.g., deep networks) was specifically to deal with the cohort size problem. LASSO has reported advantages to avoid overfitting, and thus it is an ideal method for smaller datasets. This method has also been used by others in similar studies with comparable sample sizes. A typical example is the seminal study published by Good et al in Nature Medicine 2018.

Of note, to test the stability of our findings, two additional AML patients (namely P33 and P34, treated with dose-reduced induction therapy) were added to the FlowSOM and LASSO cox regression analysis of the first 32 patients. The result of this second analysis was that p-ERK1/2 ($p=0.0019$, $p\text{-adj}=0.0038$, Log-HR 1.25) and p-p38 ($p=0.0020$, $p\text{-adj}=0.004$, Log-HR 2.07) in MC 9 at 24 hours was significant at predicting two-year survival. When analyzing five-year survival with P33 and P34 included, significance was identified in p-ERK1/2 at 24h in myeloid blasts (MC9) ($p=0.0011$, $p\text{-adj}=0.0022$, Log-HR 1.29) and p-Rb at 24 hours in the B-cells (MC5) ($p=0.003$, $p\text{-adj}=0.006$, Log-HR 1.67) (Material and methods, line 836). We acknowledge that this is a small validation set, but these analyses indicate that the results are indeed stable in a slightly expanded patient cohort.

Nonetheless, we agree with the reviewer that the cohort size is small, and even though the results might be true for this particular small cohort of AML patients it may not apply to all AML patients. The results have to be validated in future studies of larger AML cohorts. We have made the following edits to the revised manuscript in order to make the readers aware of this limitation of the study:

- i) " Chemotherapy-induced changes in intracellular signaling during the first 24 hours of treatment **may** predict long-term survival" (line 232)*
- ii) "One limitation of our study is the small cohort size, and follow-up studies with larger patient cohorts are required to further validate our findings." (line 694).*

As a final note, we would like to point out that low statistical power increases the risk of type II error, not type I error; i.e. if the analysis did not return any statistically significant results, we would not be able to claim that there were no associations. However, our results are highly significant, and therefore we believe that the observed associations are not spurious, and should be tested with the available validation approaches.

6. Second, all the p-values were without multiple comparison corrections and, as a result, the type I error (or more precisely the family wise error) could not be controlled.

6.1 We have added the adjusted p-values for all analyses involving multiple testing in the revised manuscript.

7. Third, the machine learning results were not described clearly. How did you set up your training and validation samples? Did you do cross-validation?

7.1 The machine learning approach was validated using (nested) cross-validation. As described in the manuscript: "For survival analyses, we applied a Cox Lasso regression model with automatic feature selection and nested leave-one-out cross-validation to determine the regularization parameter." (Material and methods- line 820) and main text (line 237). Since we used leave-one-out cross-validation and only had 32 patients, we could use all possible one-patient subsets as test samples, all one-patient subsets as validation samples, and all 30-patient subsets as training data.

We agree that the description of the machine learning approach should be improved. To provide a better description of the nested leave-one-out cross validation, we added a supplementary figure (Supplementary Figure 5) to the revised manuscript. We have also added some additional text to the methods section (line 821)

Supplementary Figure 5

As the reviewer points out, we have used machine learning to generate hypotheses from a complex mass cytometry dataset. We used it to dissect the data, by linking single cells to predictive markers. Our aim was to present this as a framework that can be used in future larger studies, not only for AML but possibly also other forms of cancer.

8. Normally, machine learning helps you generate hypotheses to test in future studies. I am not sure how machine learning strengthens your results here.

8.1 We do not intend to claim that machine learning strengthens our results, but it was necessary for the discovery of phospho-ERK1/2 in a putative model for prediction of survival. As pointed out by the reviewer, CyTOF data are multidimensional and complex. Thus, the downstream analysis requires effective computational methods that can handle such data. In this case, we used lasso regression - as is commonly used in the mass cytometry literature.

Of note, the hypothesis generated by the machine learning algorithm was indeed validated using other methods in this study. First, manual analysis of the mass cytometry dataset confirmed the association of phospho-ERK1/2 with survival in the disease-specific cell population. Next, PB samples from a subset of the patients were also analyzed by RNAseq, proteomics, and drug sensitivity and resistance testing. These independent analyses also indicated activation of the ERK1/2 signaling pathway in the group of patients identified as "pERK1/2 high" in the initial analysis. In our opinion, the application of several independent experimental approaches to inspect the ERK1/2 signaling pathway at both the RNA-, protein- and functional levels strengthens our findings.

Reviewers' Comments:

Reviewer #3:

Remarks to the Author:

My last question has been well answered.

Reviewer #5:

None